# Pre-Columbian earth-builders settled along the entire southern rim of the Amazon

Jonas Gregorio de Souza [1], Denise Pahl Schaan[2], Mark Robinson[1], Antonia Damasceno Barbosa[2], Luiz E. O. C. Aragão [3,4], Ben Hur Marimon Jr.[5], Beatriz Schwantes Marimon[5], Izaias Brasil da Silva[3], Salman Saeed Khan [1], Francisco Ruji Nakahara[2] & José Iriarte [1]

The discovery of large geometrical earthworks in interfluvial settings of southern Amazonia has challenged the idea that Pre-Columbian populations were concentrated along the major floodplains. However, a spatial gap in the archaeological record of the Amazon has limited the assessment of the territorial extent of earth-builders. Here, we report the discovery of Pre-Columbian ditched enclosures in the Tapajós headwaters. The results show that an 1800 km stretch of southern Amazonia was occupied by earth-building cultures living in fortified villages ~Cal AD 1250–1500. We model earthwork distribution in this broad region using recorded sites, with environmental and terrain variables as predictors, estimating that earthworks will be found over ~400,000 km$^2$ of southern Amazonia. We conclude that the interfluves and minor tributaries of southern Amazonia sustained high population densities, calling for a re-evaluation of the role of this region for Pre-Columbian cultural developments and environmental impact.

---

[1] Department of Archaeology, College of Humanities, University of Exeter, Laver Building, North Park Road, Exeter EX4 4QE, UK. [2] Department of Anthropology, Federal University of Pará, Belém 66075-110, Brazil. [3] Remote Sensing Division, National Institute for Space Research, São José dos Campos 12227-010 SP, Brazil. [4] College of Life and Environmental Sciences, University of Exeter, Exeter EX4 4RJ, UK. [5] Universidade do Estado de Mato Grosso, Campus de Nova Xavantina, Nova Xavantina, MT 78690-000, Brazil. Correspondence and requests for materials should be addressed to J.Souza. (email: J.Gregorio-De-Souza@exeter.ac.uk)

The protection of rainforests and the development of sustainable land-use practices in the tropics are of global significance as these forests represent a major reservoir of biodiversity and are crucial for the regulation of Earth's climate[1,2]. An understanding of the historical role of humans in shaping Amazonian landscapes, and to what extent these forests were resilient to historical disturbance, is critical to making informed policy decisions on sustainable futures[3,4]. However, the nature and scale of Pre-Columbian land use and its modern legacy on Amazonian landscapes is a debate that remains largely unresolved because huge swaths of the rainforest are still unexplored[3–13].

The interfluvial (*terra firme*) forests that account for ~95% of the Amazon are particularly uncharted[10,14]. These areas have been archaeologically neglected following traditional views that Pre-Columbian people concentrated on resource-rich floodplains[15,16]. However, the discovery of large Pre-Columbian earthworks in *terra firme* along the Southern Rim of the Amazon (SRA) undermines the assumption that these areas were marginal in terms of past human impact and the development of complex societies.

Earthworks occur from the Llanos de Moxos savannas in Bolivia[17,18], through the southwestern Amazonian moist forests of Acre[19,20], to the seasonal forests of the Upper Xingu in Mato Grosso[21,22] (Fig. 1). The association of ditched enclosures and large habitation mounds with canals, causeways, fish weirs, water reservoirs and raised fields, covering at least 250,000 ha of seasonally-flooded savanna[18,23–27], has led some to speak of whole domesticated landscapes in Amazonia[28] with major consequences for understanding the past human footprint in the modern forest composition[11].

In the state of Acre, Brazil, geometrically-patterned ditched enclosures known as 'geoglyphs' combine square, circular and hexagonal earthworks[29–31] (Fig. 2a, b). They are interpreted as ceremonial centres based on votive findings and lack of occupation debris (Supplementary Note 1). Mounded ring villages are often found in the vicinity or inside geoglyphs (Fig. 2c), and are potentially ancestral to the ethnographic ring villages of Arawak-speakers found along the SRA (Supplementary Note 1). In the Llanos de Moxos, enclosures known as 'ring ditches' are found in forest islands of the flooded savanna associated with water-management earthworks such as canals[26,32,33]. Ring ditches are probably the archaeological correlates of palisaded villages described in historical accounts, with many of the sites containing domestic debris amidst other features, such as multiple burials (Supplementary Note 1). Finally, in the Upper Xingu, ~900 km to the east, large fortified settlements, sometimes enclosed by multiple ditches, have been described as representing a form of Pre-Columbian low-density urbanism[21,22,34]. They are connected through a network of causeways, resembling the modern organisation of indigenous communities in the Upper Xingu, except on a much larger scale (Supplementary Note 1).

It was long suspected that the geographically widespread enclosures found from Acre to the Upper Xingu were related phenomena, but the potential connection between these earth-building cultures remained unresolved due to a dearth of

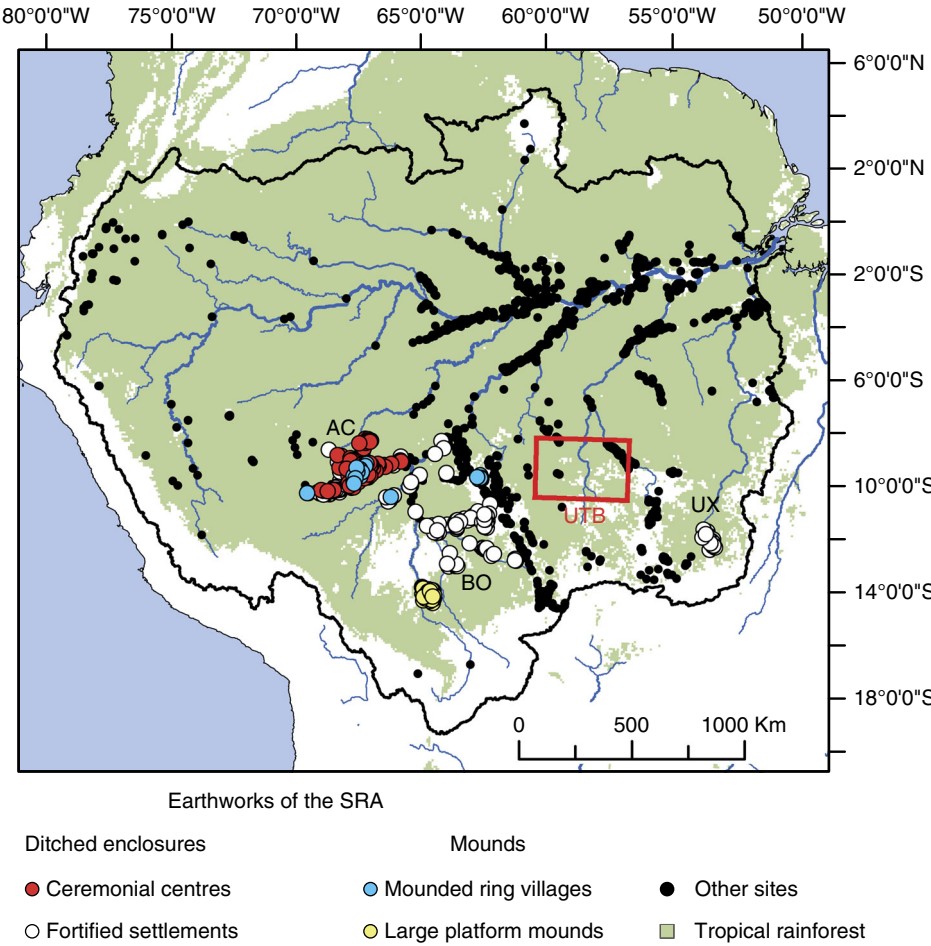

**Fig. 1** Distribution of earthworks in the SRA and other sites in the Amazon[68,69]. Different categories of earthworks as per Supplementary Note 1. Sub-regions of the SRA discussed in the text: AC Geoglyphs of Acre, BO Ring ditches of Bolivia, UX Upper Xingu, UTB survey area in the Upper Tapajós Basin (the area delimited by the red box is shown in detail in Fig. 4). Scale bar = 1000 km

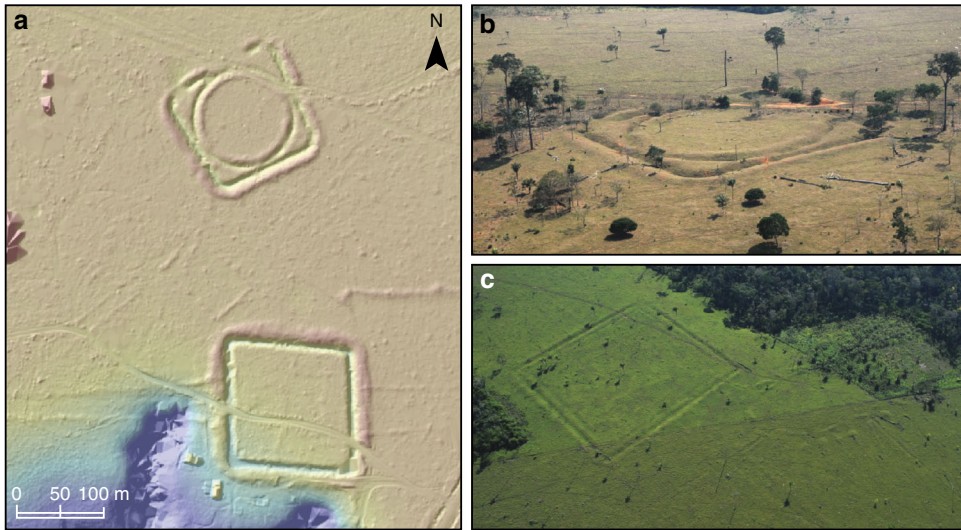

**Fig. 2** Geoglyphs and mounded ring villages. **a** LiDAR digital terrain model of the Jacó Sá site showing geometric ditched enclosures, walled enclosures and avenues. Scale bar = 100 m. **b** Aerial photo of one of the structures at Jacó Sá site. **c** Aerial photo of Fonte Boa site showing a mounded ring village with radiating roads (right) built next to an earlier geometric enclosure (left). Aerial photographs are part of the collection of CNPq research group Geoglyphs of Western Amazonia directed by Denise Schaan

archaeological data in the bridging areas[35]. Ethnohistoric evidence speaks in favour of such connections. Historically, the southern periphery of the Amazon was occupied by a chain of Arawak-speaking populations[36,37]. They shared a common ethos of settled village life, ranked social organisations, macro-regional integration and inter-community exchange[34,37–39]. At the same time, regions such as southwestern Amazonia were home to one of the highest diversities of language families within Amazonia, and the multi-ethnic/multilinguistic nature of regional systems is exemplified by the Upper Xingu[34]. The belt of Arawak and other groups along the SRA has been hypothesised to constitute a formative supra-regional system that was present from late Pre-Columbian times[37]. If true, this connection would suggest an uninterrupted distribution of earthworks along 1800 km east–west in the SRA and a more intense Pre-Columbian human impact in the forests of this region than previously postulated. However, a major gap in the archaeology of the Upper Tapajós Basin (UTB) hampered the evaluation of this hypothesis.

That the UTB hid settlements comparable to those found to the east and west was suggested by 18th century accounts, where the region was portrayed as densely populated, with large villages connected by straight and wide roads[40]. In this paper, we present new remote sensing, survey and excavation data from the previously uncharted UTB. We report the discovery of 81 new sites, filling a gap in the archaeology of the Amazon. The discovery shows that there are no discontinuities in the spatial and temporal distribution of earthworks, spanning 1800 km east–west (Fig. 1), changing traditional views about the scale of Pre-Columbian landscape modification and sustainability of past practises in the transitional forests of the SRA.

## Results

**General characteristics of the surveyed sites**. The survey in the UTB located 81 Pre-Columbian sites, some of which combined multiple structures, for a total of 104 earthworks (Supplementary Table 4). Archaeological sites are mostly found on relatively flat landscapes with gentle hills in elevations between 100–300 m. Areas of residual plateaus and *serras* (mountain chains) are generally devoid of earthworks. Ditched enclosures of varying diameters (11–363 m), layouts, and containing different types and numbers of

associated earthworks (such as mounds and avenues) are situated on the tops of small plateaus, overlooking rivers and streams (Fig. 3a). Similar to other fortified villages of SRA[21,22], a majority of them contain superficial ceramics and Amazonian Dark Earth (ADE), confirming the presence of past human occupation. The formation of ADE further suggests long-term occupation of the sites.

The smallest enclosures, with average diameters of ~30 m (Fig. 3b), are clustered between the Juruena and Aripuanã rivers in the western half of the research area (Fig. 4). Small ditched enclosures are frequent, representing 30% of our sample, and tend to appear isolated. In a few cases they are part of compounds, either with other small structures or in the vicinity of much larger earthworks. The ditches are circular and surrounded by an outer bank, often with interruptions suggestive of an entrance. Areas of ADE with ceramics and polished stone axes are found at a distance of ~80–100 m from the small enclosures. These sites are reminiscent of the Eastern Complex defined for the Upper Xingu[34] comprising circular enclosures with small dimensions (20–40 m diameter) and openings. Excavations in the Eastern Complex sites revealed domestic features inside the enclosures and middens with ADE in their exterior. They are interpreted by Heckenberger[34] as single large communal houses for extended families or upper-status individuals. In the UTB, small enclosures are often found inside or attached to larger ditches, forming compounds (Fig. 3c).

The layouts of the ditches surrounding fortified villages in the UTB are not perfectly circular, but irregular and often wavy, as if built in several independent segments (Fig. 3d). The largest fortified settlements are located in the eastern half of the research area, between the Juruena and the Teles Pires rivers (Fig. 4). Many of the larger sites, whose layout initially appeared to be irregular, in fact tend to repeat a roughly hexagonal shape, often with a pointed end, suggesting a certain degree of planning and uniformity in their construction. The ditches of the sites surveyed in the UTB vary from 1 to 3 m deep with 1 m high internal and external embankments except in the smallest enclosures where walls are only found in the exterior of the ditch.

Some of the largest sites in the UTB are distinguished not only by their dimensions, but also by their elaborate architecture that combines multiple earthworks, potentially reflecting long histories of construction and remodelling. Mt-07-Boa Vista is an enclosure 390 m in diameter built on a small plateau overlooking

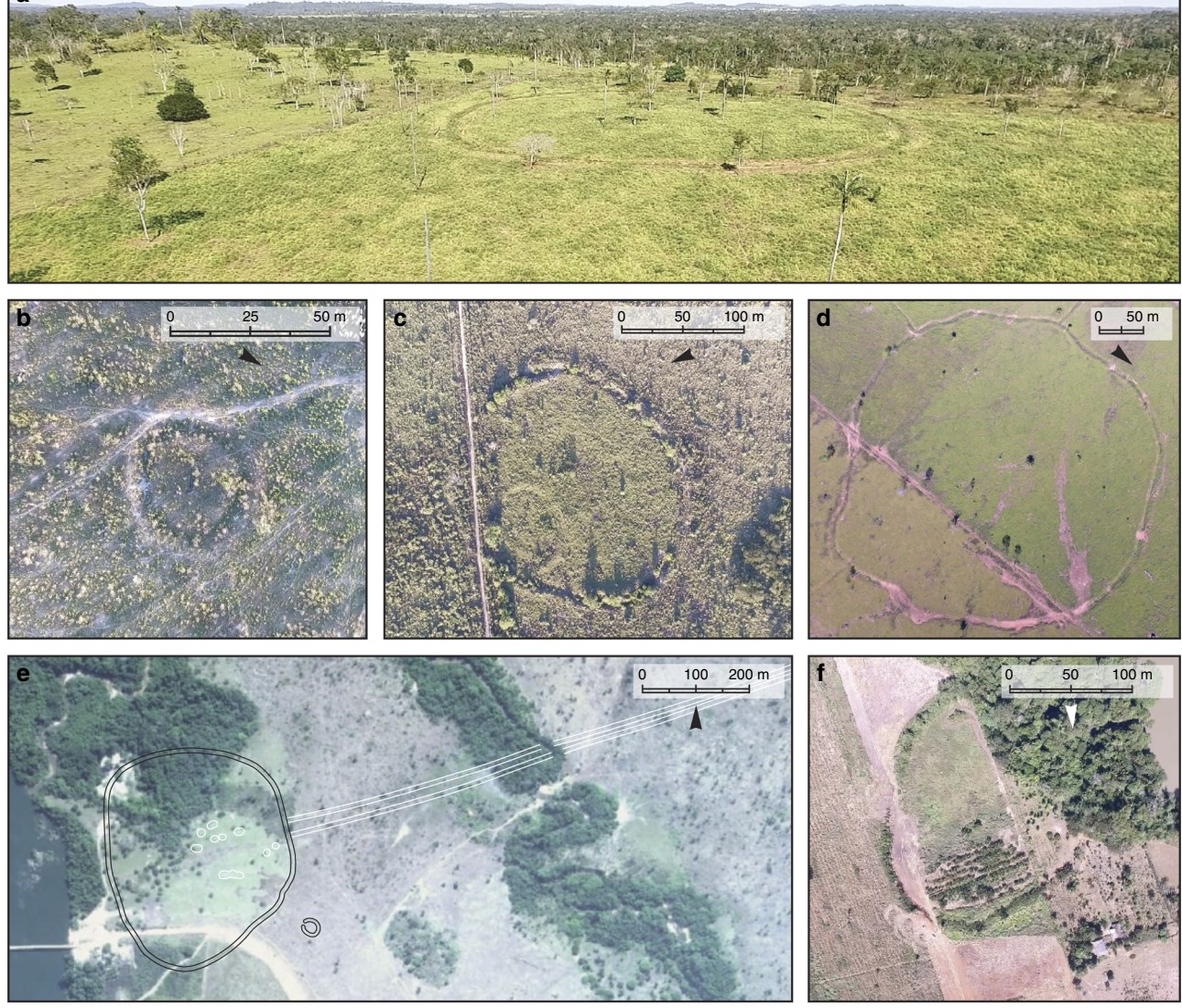

**Fig. 3** Ditched enclosures of the UTB. **a** Typical position of ditched enclosures in the landscape, on plateaus overlooking rivers (site Mt-05); **b** Example of a small enclosure (site Z-Mt-29). Scale bar = 50 m; **c** Compound structure with a small enclosure in the interior of a larger one (site Z-Mt-05). Scale bar = 100 m; **d** Part of the site Z-Mt-04, showing the wavy pattern of ditch construction. Scale bar = 50 m; **e** Plan of a major regional centre, site Mt-07, with ditches outlined in black and mounds/walls outlined in white (Satellite image ©2018 Google, DigitalGlobe). Scale bar = 200 m; **f** Site Mt-04. Scale bar = 100 m

the Aripuanã River, ~80 m from its margins. It has 11 mounds circularly arranged at the centre of the enclosure. A sunken road framed by two linear earthen banks exits the ditch to the east towards a small stream, continuing in a straight line for 1.4 km, while a small enclosure (20 m diameter) is located 50 m southeast of the main structure (Fig. 3e).

In addition to ditched enclosures, five mounded ring villages have been discovered in the UTB, primarily restricted to a small area west of the Juruena River (Fig. 4). Mounds are arranged around circular plazas ~100–120 m diameter, with sunken roads radiating in many directions and stretching up to 1.4 km. In one case the mounded village has been built inside a large ditched enclosure (Fig. 3e), and in two other cases small enclosures can be found ~100 m away.

**Settlement patterns in the Upper Tapajós Basin.** In order to explore settlement patterns in the UTB, we classified sites according to the total area into individual hamlets and small villages (up to ~2 ha), medium to large settlements (~2–5 ha), and major regional centres (over 5 ha) (see Methods, Supplementary Fig. 1). A single

site, Z-Mt-04 (Fig. 3d), overshadows all others with an area of ~19 ha, due to its combining two of the largest enclosures in the same compound, and may constitute a category of its own.

The spatial distribution of sites shows that the smallest enclosures are significantly clustered, whereas the largest sites are regularly spaced. Distance between medium/large sites ranged from 24.8 to 33.17 km, although this was not found to be significant. On the other hand, the first-order settlements in the site hierarchy were separated in average by 85.5 km in a significant regularly-spaced pattern (see Methods, Supplementary Fig. 2). We interpret the clusters of small enclosures gravitating around the largest sites as reflecting independent settlements with catchments of ~700 km² nested within peer-polity territories of ~5000 km² (Fig. 4). We argue that settlement patterns in the UTB point to regionally integrated polities analogous to the 'galactic' system of the Upper Xingu[21,22]. Under such organisation, larger and architecturally more elaborated sites – 'exemplary centres' – constitute population nodes exerting influence over surrounding satellite villages. Among modern indigenous societies of the Upper Xingu, communities are independent, but informal

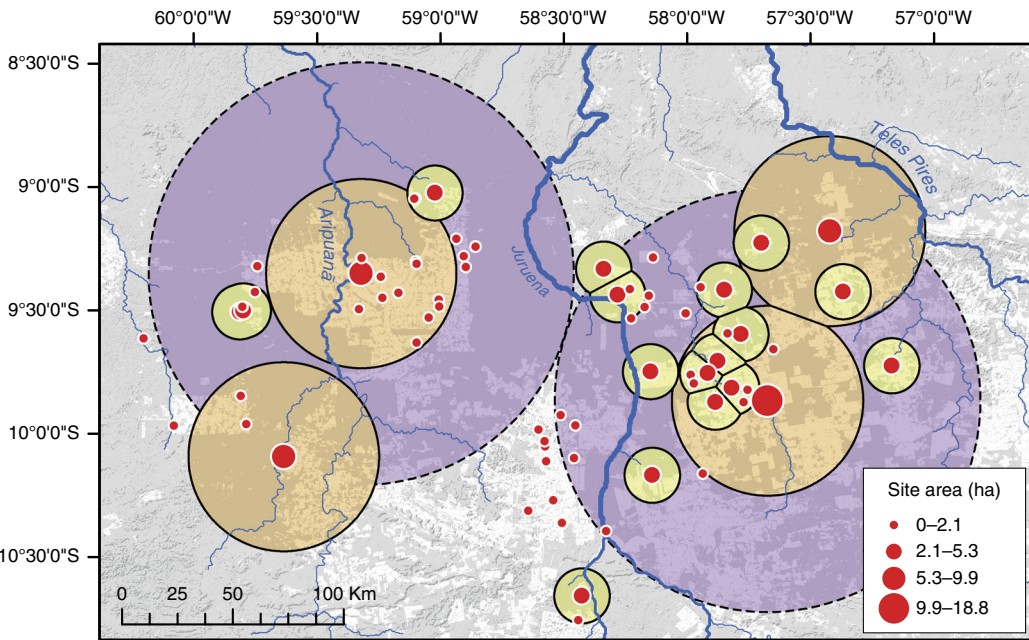

**Fig. 4** Distribution of ditched enclosures in the surveyed area of the UTB. Another four sites have been recorded ~150–200 km to the west and east and are not shown. Small circles are buffers of ~25 km, corresponding to the territory of medium and large sites based on the average distance between them. The large circles around the major centres are buffers of ~85 km based on the average distance between sites of this class. Dashed circles highlight the potential western and eastern regional site systems. Background landsat imagery from 2017 showing forests (grey) and deforested patches (white). Landsat-7 image courtesy of the U.S. Geological Survey. For location of the study area in southern Amazonia, see red box in Fig. 1

**Table 1 Radiocarbon dates from site Mt-04**

| Unit | Level | Context | Conventional $^{14}$C date | Laboratory number | $\delta^{13}$C ‰ | Cal. AD 2σ |
|---|---|---|---|---|---|---|
| Test unit 1 | 40–50 cm | Base of ADE inside enclosure | 520 ± 30 | Beta-469161 | −27.6 | 1410–1455 |
| Test unit 2 | 40–50 cm | Base of ADE inside enclosure | 500 ± 30 | Beta-469164 | −25.8 | 1415–1460 |
| Test unit 3 | 10–20 cm | Base of ADE outside enclosure | 520 ± 30 | Beta-469165 | −26.9 | 1410–1455 |

hierarchies have been reported, with the largest villages holding higher prestige and ritual importance[34]. Moreover, temporary asymmetries have been described, linked to events such as the hosting of intercommunity rituals[34]. At focal points in the west and east of our study area, the two largest sites so far recorded (Boa Vista and Z-Mt04, separated by ~190 km) may have functioned as 'exemplary centres' in their respective regions. The architecture of sites such as Boa Vista (Fig. 3e) is reminiscent of the plaza centres of the Upper Xingu and suggests that it might have incorporated additional ceremonial and political functions beyond operating as a major population hub[22,34,37]. Absent from the UTB, however, is the network of roads reported for the Upper Xingu, which are so far unique to that area[22,34,37].

Beyond the UTB and the Upper Xingu, a regional hierarchy of sites has also been proposed to explain the distribution of large platform mounds in the Bolivian Llanos de Moxos[24]. The geoglyphs of Acre, however, were shown not to conform to any significant distribution in terms of site size[20], and no such patterns are reported for the ring ditches of Bolivia[26]. Thus, distinct social formations might have existed along the 1800 km stretch of the SRA occupied by earth builders, in line with variability in ceramic styles and architectural traditions.

**Cultural contents and chronology of the sites**. In order to examine the cultural contents, stratigraphy and chronology of the UTB sites and how they are compared to other ditched enclosures of the SRA, we have selected site Mt-04 to conduct excavations

(Fig. 3f, Supplementary Fig. 3). The interior of the site contained a thick layer of ADE 50 cm deep, which was only 20 cm deep outside of the ditch. The ADE contained abundant ceramics and charcoal and was superimposed to a layer of extensive burning at the transition with the local oxisols. The basal burning and pre-ADE strata were devoid of archaeological remains. The presence of ADE and high density of ceramics in the area surrounded by the enclosure reinforce the domestic nature and permanent occupation of the site. Radiocarbon dates were obtained for wood charcoal in association with ceramics near the base of the ADE in each of the test pits. All dates fall between Cal AD 1410 and 1460 (Table 1). The dates obtained for Mt-04 show contemporaneity with the construction of ditched enclosures elsewhere in the SRA. A compilation of the available radiocarbon dates (Supplementary Tables 1–3) shows a peak of activity ~Cal AD 1250–1500 both in Bolivia and the Upper Xingu, with mounded ring villages also emerging during this period (Fig. 5). The geoglyphs of Acre, however, flourished earlier than most other earthworks along the SRA, ~Cal AD 100–400, with a clear demise after Cal AD 1000 (Fig. 5).

Ceramics recovered from Mt-04 exhibit a myriad of decorative styles (Supplementary Fig. 4) and are distinct from those reported for the enclosures of Acre[30], Bolivia[26] and the Upper Xingu[34], showing that each of these regions was occupied by a particular ceramic tradition, all sharing the practise of building ditched enclosures.

**Predicting enclosure distribution in southern Amazonia**. We modelled the potential full geographical extent of ditched

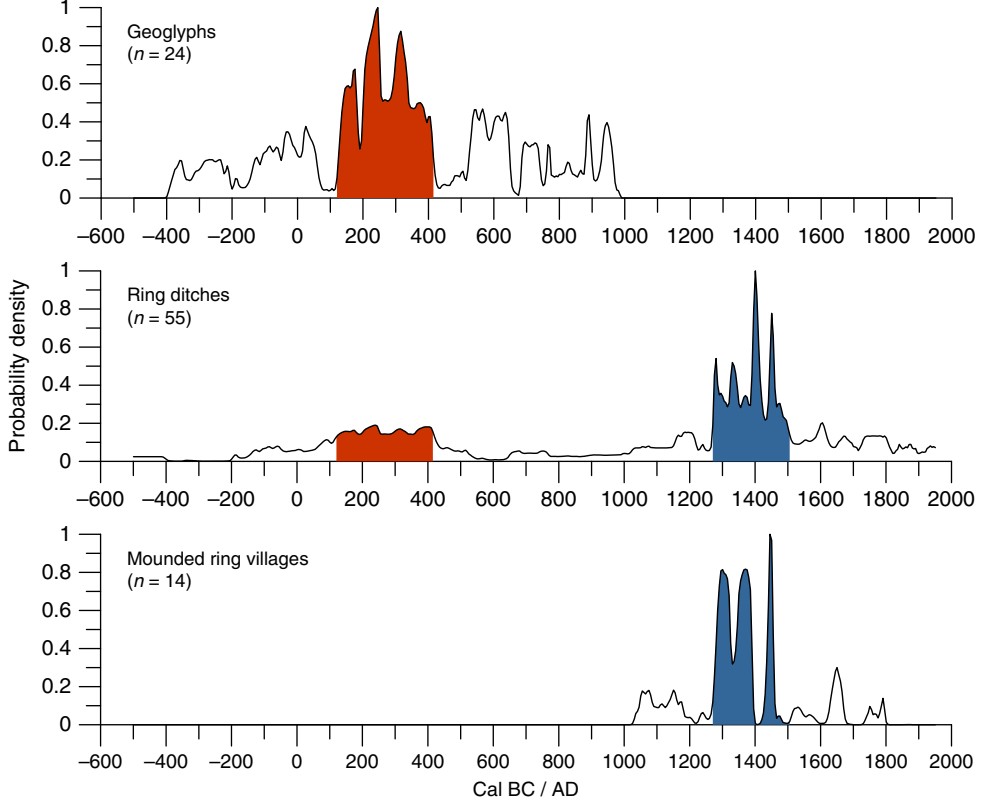

**Fig. 5** Sum of the probability distributions (SPDs) of radiocarbon dates for earthworks in the SRA. Curves are shown for different categories of earthworks. Peaks in site construction and occupation are highlighted for geoglyphs/early ring ditches (red) and later ring ditches/mounded ring villages (blue). *n* number of dates

enclosures in the SRA using maximum entropy algorithms[41–43] based on archaeological site coordinates (previously recorded ditched enclosures and new sites surveyed in the UTB) and environmental parameters, including 11 bioclimatic, eight edaphic and four terrain variables. Maximum entropy modelling, primarily used in biology for predicting species distribution based on presence only data, has been recently employed to predict the distribution of ADE[44], geoglyphs[45] other archaeological sites and maize agriculture in the Amazon[46]. Here, we expand those applications to the prediction of ditched enclosures across the whole SRA (Fig. 6).

Bioclimatic variables related to temperature and precipitation seasonality were found to be the most relevant factors in predicting site distribution (Table 2, Supplementary Figs. 8–10). In the evaluation of variable contribution, the bioclimatic parameters with highest percent contribution and permutation importance were precipitation of coldest quarter, precipitation of driest quarter and temperature seasonality (Table 2). Annual precipitation, temperature of the warmest month and temperature annual range are the next most significant bioclimatic parameters. Ditched enclosures are found in areas with pronounced seasonality in rainfall, ranging from 700 to 1000 mm during the wettest quarter to 50–120 mm during the driest quarter (Supplementary Fig. 8). Temperature variation is also high, with a difference between the absolute maximum and minimum absolute temperatures over the year reaching 14–24 °C (Supplementary Fig. 8). The terrain variable with the highest significance was elevation, confirming the observation that ditched enclosures tend to be located in relatively high terrain (140–320 m) (Supplementary Fig. 10). Among the terrain predictors, distance to rivers was the second most significant,

but it had very low percent contribution and permutation importance when compared to the bioclimatic variables (Table 2).

Our modelled distribution of ditched enclosures is remarkably distinct from that of ADE sites reported by McMichael et al.[44]. Bioclimatic factors are less important for the prediction of the latter, and terrain parameters are fundamentally different. ADE sites are located at elevations below 100 m and within 10 km of perennial rivers[44]. In contrast, the SRA enclosures are located in higher terrain (140–320 m), and distance from rivers, although still a significant terrain predictor, was found to be much less important, with earthworks located up to 40 km from navigable rivers (Table 2, Supplementary Fig. 10). The relationship between earthworks and proximity to rivers, however, varies according to region. Confirming field observations[26,29,34], enclosures in Bolivia and the Upper Xingu are predominantly found within 10 km of rivers, whereas in Acre and the Upper Tapajós sites are as likely to be found at longer distances (Fig. 7). Overall, our data confirm that the earthworks of the SRA represent a distinct settlement strategy relative to the ADE sites of the major river floodplains[44].

In order to estimate the total number of geometrical enclosures in the SRA, we projected the site density in the UTB to the whole area predicted by the model to contain earthworks. The predicted area in ten iterations of the model varied from 355,690 to 493,377 km², with an average of 434,860 km² (see Methods). Earthwork density in the UTB—considering only the extent of the deforested areas that were surveyed in satellite imagery (20,019 km²)—is of 0.003 sites/km². When projected to the average total area predicted by MaxEnt, this amounts to 1304 sites. Because site density is reportedly higher in other regions of the SRA, such as the Upper Xingu and Acre, this number is likely to be an underestimation.

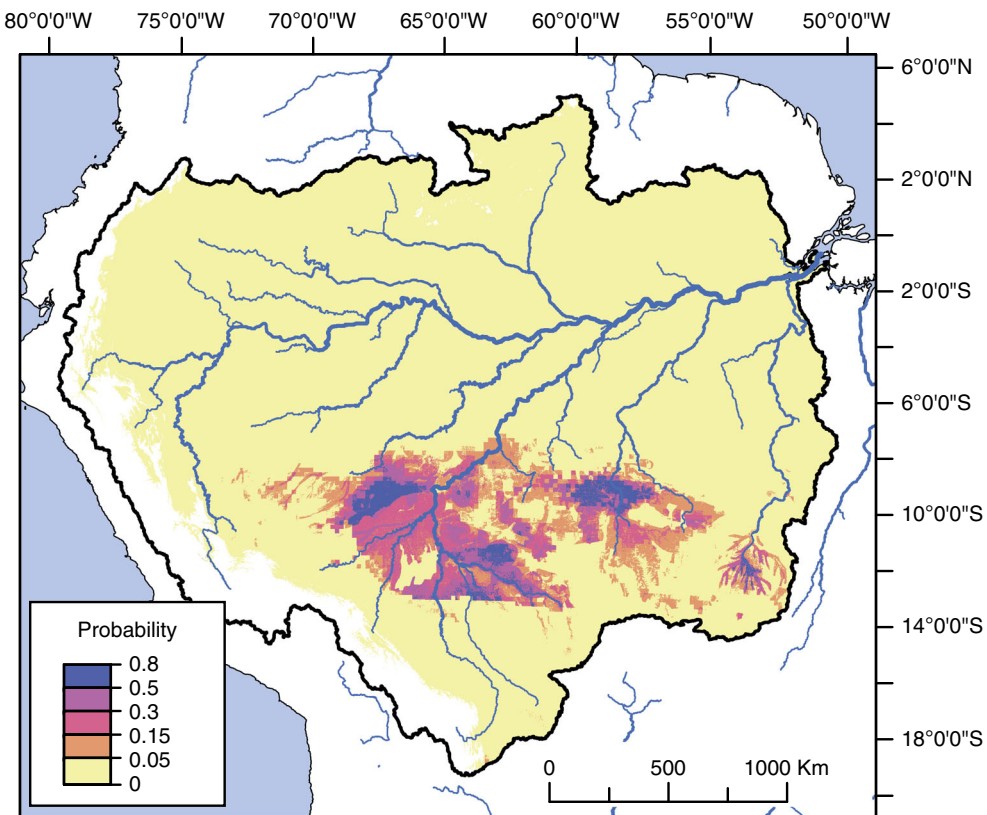

**Fig. 6** Maximum entropy (MaxEnt) model of ditched enclosure distribution. The probability scale is given in the default logistic output of MaxEnt

We further used the results of the model to estimate the population of the SRA during late Pre-Columbian times (see Methods, Supplementary Table 4). We derived population figures for the UTB based on site area and ethnographically-derived formulae[47] and projected the results to the total area predicted by the model to contain earthworks. Assuming site contemporaneity, the total population calculated for the UTB sites was 21,340 with a density of 1.06 persons/km². When this number is projected to the total area of potential occurrence of ditched enclosures in the SRA, the result is a population of 463,555. This is almost certainly an underestimation, given the higher population densities estimated for other areas of the SRA at the eve of the Columbian encounter. Population counts for the Llanos de Moxos in the 17th century based on Jesuit records vary from 40,000 to over 100,000[32,48], whereas in the Upper Xingu it was reported to be only 3000–4000 individuals[34]. These numbers reflect the impact of European diseases, with Pre-Columbian population estimates based on settlement area and extent of ridged fields of 350,000 and 50,000, respectively[21,48]. In both cases, this translates to a density of ~2.5 persons/km². When projected to the total area of the SRA predicted to contain earthworks, the result is 1,087,150 inhabitants. Therefore, we assume that a population between ~500 thousand and 1 million for the SRA in late Pre-Columbian times (Cal AD 1250–1500) is a reasonable estimate.

## Discussion

The possible geographical continuity of late Pre-Columbian earth building cultures across all southern Amazonia has been envisaged for nearly a decade[35]; however, archaeological evidence of such long-range connections remained elusive. The results of our survey in the Tapajós headwaters confirm that ditched enclosures are found over an 1800 km east–west belt along the SRA. The expansion in earthwork construction and the peak of occupation

at most sites—both in the eastern and western extremes of the SRA—occurred between Cal AD 1250 and 1500. During this time, the whole SRA was settled by dense populations living in fortified villages, some of which (e.g., the Upper Tapajós, the Upper Xingu) appear to have been organised in regionally-integrated peer-polity systems.

The results of our predictive model of ditched enclosures show that, despite the enormous distances covered, earthworks are found across areas of notable environmental similarity, with pronounced seasonality in rainfall and temperature. Our results are in conformity with the suggestion that Pre-Columbian populations were denser and their environmental impact, including earthwork construction, was higher in seasonal climates such as found in the SRA[49–51]. The seasonal drought of the transitional forests of this region probably facilitated clearance for the construction of earthworks. The easy-to-clear vegetation and the more fertile/less weathered soils of seasonally-dry forests are factors that made them attractive to Pre-Columbian farmers, and the ancestors of many Neotropical crops grew in seasonally-dry forests[52]. In particular, the belt of transitional forests along the SRA has been shown to be high in relative richness of domesticated species including *Bertholletia excelsa* (Brazil nut)[7,11,53].

The results of the model allow us to predict that ditched enclosures will be found over an area of ~400,000 km² along the whole SRA. We estimate that ~1000–1500 enclosures exist in the southern periphery of the Amazon, implying that up to about two-thirds of the sites are yet to be found. We infer a regional population between ~500,000 and 1 million for the SRA during Late Pre-Columbian times, based on a bottom-up approach using empirical archaeological data. This figure calls for a re-evaluation of models solely based on carrying capacity or projections of recent ethnographic data, which often presuppose that most of the Amazonian Pre-Columbian population was settled along riverbank settings, with low densities and limited environmental

**Table 2 Percent contribution and permutation importance of the bioclimatic, edaphic and terrain variables included in the final MaxEnt model**

| Variable | Percent contribution | Permutation importance |
|---|---|---|
| Precipitation of coldest quarter | 32.2 | 3 |
| Precipitation of driest quarter | 13.9 | 9.5 |
| Temperature seasonality | 11.7 | 26.1 |
| Cation exchange capacity | 9.5 | 2.1 |
| Elevation | 7.7 | 3.5 |
| Annual precipitation | 5.3 | 4.1 |
| Max. temperature of warmest month | 5 | 8.1 |
| Cation exchange capacity | 2.6 | 0.8 |
| Temperature annual range | 2.5 | 22.1 |
| Isothermality | 1.8 | 0.4 |
| Precipitation of wettest quarter | 1.5 | 1.6 |
| Gravel content | 1.4 | 0.5 |
| Distance to rivers | 1.4 | 2.2 |
| Precipitation of warmest quarter | 0.9 | 1.2 |
| Sand fraction | 0.6 | 8.5 |
| Silt fraction | 0.5 | 0.4 |
| Slope | 0.5 | 1 |
| Annual mean temperature | 0.4 | 0.6 |
| Organic carbon | 0.2 | 2.2 |
| Bulk density | 0.1 | 1.5 |
| pH | 0.1 | 0 |
| Precipitation seasonality | 0 | 0.3 |
| Topographic position index | 0 | 0 |

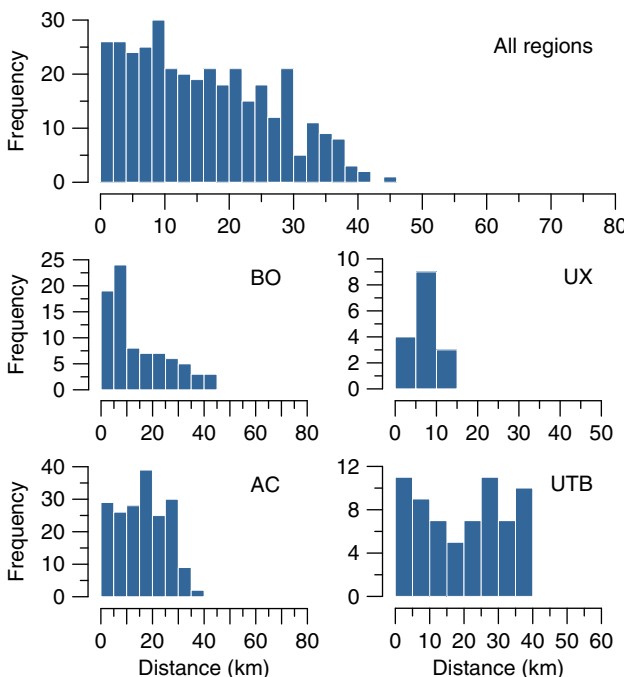

**Fig. 7** Frequency of SRA ditched enclosures as a function of distance from rivers. Perennial rivers are defined as flow accumulation >15,000 (see Methods). Regions: BO ring ditches of Bolivia and neighbouring Rondônia, UX Upper Xingu, AC geoglyphs of Acre and neighbouring Amazonas, UTB Upper Tapajós Basin. The limits of the x axis correspond to the maximum potential distance in the respective regions

impact in *terra firme* areas[7,51,54]. Moreover, the fact that the predicted area of earthworks along the SRA, an area comprising merely 7% of the Amazon, could have sustained a population in the hundreds of thousands (even if one accepts the lower threshold of our estimate), definitively discredits early low estimates of 1.5–2 million inhabitants for the whole basin[55].

The sheer distances involved in the territorial extent of earth builders imply a diversity of traditions and socio-political trajectories for each sub-region of the SRA. This is witnessed by the idiosyncrasies in architecture, ceramic styles and settlement patterns in Acre, Bolivia, and the Upper Xingu, to which the new findings from the UTB must be added. Our results further reinforce the significance of seasonal transitional forests for cultural developments in the Amazon basin. Finally, we call into question the prevalent idea that major waterways were the main communication routes in Amazonia and stress the importance of parallel networks of complex societies and formative supra-regional systems[37] established along interfluves and minor tributaries of the periphery of the Amazon basin.

## Methods

**Satellite imagery and ground-truthing**. We designed a multi-staged research programme in the selected area of the UTB involving systematic survey of satellite imagery, ground checking of a sample of structures and test excavations. Deforested areas in the basins of the Juruena and Teles Pires rivers (the headwaters of the Tapajós) and immediate surroundings (Figs. 1 and 4) were systematically surveyed using free satellite imagery available in Google Earth v 7.1.8.3036 and http://zoom.earth, covering an area of ~54,000 km². The surveyed area encompasses 12 municipalities in the northwest of the state of Mato Grosso, Brazil. Most of the area has submetre resolution coverage from DigitalGlobe, allowing even the smallest earthworks to be identified. Sites were chosen for ground-truthing based on their size, layout and type of earthworks present, so as to constitute a reliable sample. The combination of systematic analysis of satellite imagery with ground survey to confirm archaeological sites has been applied successfully elsewhere in the SRA[56]. We visited 24 sites consisting of small, medium and large enclosures, compounds with multiple enclosures, and circular mounded villages. All visited locations were confirmed as real archaeological sites and are considered to be representative of the region as a whole.

**Site-size distribution**. In order to explore settlement patterns and potential site hierarchies in the Tapajós headwaters, we classified sites according to area. We considered individual enclosure area and total site area for those with multiple structures. Using Jenks natural breaks method in R v. 3.3.3, we initially devised a threefold classification based on proposals for neighbouring regions[21], but found that a fourfold classification produced a better fit (Supplementary Fig. 1). Sites were divided into individual hamlets and small villages (less than 2.11 ha), medium to large settlements (2.11–5.26 ha), major regional centres (5.26–9.88 ha) and site Mt-04 as an outlier (18.79 ha) (Supplementary Fig. 1). The spatial distribution of sites was analysed in ArcGIS 10.5 using the average nearest neighbour tool. The data points were subdivided according to different size classifications (Supplementary

Fig. 1). For each category, the average distance to the nearest neighbour and the significance of the distribution (as clustered or dispersed) were calculated using the nearest neighbour ratio (Supplementary Fig. 2).

**Excavations at site Mt-04**. Mt-04 is an enclosure of irregular shape, approximately elliptical, ~175 m along the longest axis (Fig. 3f). The ditch is 8–10 m wide and 2 m deep, with external and internal embankments 1 m high. The interior and immediate adjacencies of the enclosure contain ADE with abundant ceramics on the surface. The area is currently used for growing a variety of crops due to the high fertility of the ADE. Two 1 × 1 m test pits were excavated in the interior of the enclosure ~10 m from the ditch, while another one was placed at the same distance from its exterior (Supplementary Fig. 3). Artefacts and features were recorded in artificial 10 cm levels while taking note of the natural stratigraphy, and charcoal from secure contexts was collected for radiocarbon dating.

**Modelling earthwork distribution**. We used the default parameters of MaxEnt v. 3.3.3 (http://www.cs.princeton.edu/~schapire/maxent/) for predicting the distribution of sites. Ditched enclosure coordinates were compiled from published literature, unpublished theses and field reports, to which we added the newly discovered sites of the UTB, totalling 665 occurrence points. For environmental parameters, we used the 19 bioclimatic variables from WorldClim[57] (http://www.worldclim.org/bioclim), 16 edaphic variables taken from the Harmonised World Soil Database[58] and five terrain variables: elevation, slope, ruggedness, topographic position index (TPI) and distance from rivers (Supplementary Table 5). Elevation was obtained from the Shuttle Radar Topography Mission (https://eros.usgs.gov/). Slope, ruggedness and TPI were derived using Spatial Analyst and Land Facet Corridor Designer tools for ArcGIS 10.5 and GRASS 7.2. Distance from rivers was generated in ArcGIS using the data from HydroSHEDS[59] (http://www.hydrosheds.org) and a flow accumulation threshold of 15,000 to define perennial rivers[44]. All layers were resampled, when necessary, to match the resolution of the bioclimatic rasters (30 arc seconds) and cropped to the extent of Amazonia, defined as all regions draining into the Amazon River system, but excluding areas that are higher than 1000 m elevation, which had terrain and bioclimatic characteristics too different from the lowlands and could skew the prediction of the model (Fig. 6).

We tested for correlation between the environmental, soil and terrain layers using the 'raster' package for R v. 3.3.3 and removed parameters that were highly correlated with others of easier interpretation (Supplementary Figs. 5–7). Bioclimatic variables retained for the final model were (1) annual mean temperature; (2) isothermality; (3) temperature seasonality; (4) maximum temperature of the warmest month; (5) temperature annual range; (6) annual precipitation; (7) precipitation seasonality; (8) precipitation of wettest quarter; (9) precipitation of driest quarter; (10) precipitation of warmest quarter; and (11) precipitation of coldest quarter. In the case of the edaphic variables, we selected mainly for physical properties (which might affect earthwork construction) but also quality indicators (potentially influencing population distribution). Variables retained for the final model, all pertaining to subsoil attributes, were (1) gravel content; (2) sand fraction; (3) silt fraction; (4) clay fraction; (5) bulk density; (6) organic carbon; (7) pH; and (8) cation exchange capacity. Terrain variables retained for the final model were (1) elevation; (2) slope; (3) topographic position index; and (4) distance from rivers.

Because predictive models based on presence-only data are heavily affected by biases in sampling and spatial autocorrelation[60], and archaeological site distribution is frequently biased towards regions with more intense survey, we decided to apply measures of spatial filtering to our occurrence dataset. Spatial filtering of occurrence records when these are biased to certain locations has been shown to be more effective in reducing omission and commission errors than manipulation of background points in MaxEnt[60]. The Brazilian geoglyphs of Acre were overrepresented, as this area contained nearly four times more sites than the remainder of the SRA put together (n = 185). To reduce sampling bias, we randomly selected 185 sites from the Acre dataset for inclusion in the model. Another source of bias is the arbitrary definition of archaeological sites, since complexes of earthworks occurring within a few kilometres of each other may sometimes have been recorded as separate sites. To account for that, we further reduced the dataset by considering sites occurring within a specified distance of each other as single points. We tested radii of 2.5 and 5 km. Predictive models produced with these filtered datasets were nearly identical to one produced with the full dataset. As a compromise, we ran the final model using the dataset filtered with the 2.5 km threshold (n sites = 313). In order to assess the average behaviour of the model, we performed ten partitions of the occurrence dataset, each time randomly selecting 75% of the sites as training sample and 25% as test sample (Supplementary Fig 11). All models had areas under the ROC curve (AUC) > 0.96 showing that they performed significantly better than random in predicting site occurrence. Response curves and jackknife tests were used to measure the importance of the predictor variables. The default logistic output of MaxEnt is a surface of continuous probability values that can be converted into a binary surface of presence/absence based on several possible thresholds. In order to calculate the total potential area of earthwork occurrence, we used maximum training sensitivity plus specificity as a threshold[61]. This represents a compromise between omission errors or false negatives (failing to predict areas of site occurrence) and commission errors or false positives (predicting areas beyond actual site distribution), since the latter cannot be evaluated using presence-only data. Under this threshold, test omission rates never exceeded 16%.

**Site density and population estimates**. Population densities for the sites in the UTB were calculated based on site area and the ethnography-derived formulae of Curet[47]. Among the formulae presented by Curet, we found that the linear equation for all site types produced the most realistic results, with estimates of 20–30 people for individual hamlets and 1000–2500 dwellers for the largest settlements. The latter is in agreement with the estimates for other first-order archaeological settlements[21] as well as historical accounts of Central Brazilian ring villages with populations in the low thousands[62].

**Regional chronologies of the SRA**. SPDs were built in OxCal 4.2 using the Sum function and the ShCal13 calibration curve[63,64]. In order to account for oversampling of some sites and phases within these sites, we applied a binning procedure[65–67]. Dates within sites were ordered and those occurring within 100 years of each other were grouped into bins and merged with the R_combine function. This procedure is necessary because a sum of the calibrated dates assumes that observations are independent, whereas this is not the case when multiple dates were obtained for single sites or phases within them. The final curves for each site category were normalised to facilitate comparison.

**Data availability**. All data used in this study are available from the authors on reasonable request.

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

## Acknowledgements

The research was funded in part by the National Geographic Society (HJ-R008-17) to J.I., L.A., D.P.S., M.R., S.S.K. and J.G.S and the European Research Council project 'Pre-Columbian Amazon-Scale Transformations' (ERC-CoG 616179) to J.I. Research and was authorised by the Brazilian Institute of Heritage (IPHAN) (Permit number 01425.002432/2017-1).

## Author contributions

J.I., J.G.S., M.R. and D.P.S. designed the research; F.R.N. identified earthworks in satellite images; J.G.S., J.I., A.D.B., and I.B.S. carried out archaeological ground-truthing and excavations; J.G.S. carried out analysis of the data; J.G.S., J.I. and M.R. led the writing of the paper with inputs from all other authors.

## Additional information

**Competing interests:** The authors declare no competing interests.

