## [Peer Review File(PDF 811 kb) · Nature Communications]

Reviewers' comments:

Reviewer #1 (Remarks to the Author):

Souza et al. present new data on 86 earthworks from 67 sites identified along the southern Amazonian rim in the headwaters of the Tapajos River, model their new data alongside the locations of previously reported earthworks, and "conclude that population densities were higher than previously thought along the seasonal transitional forests of the interfluvial areas and minor tributaries of the southern Amazonian periphery". These data are a substantial contribution to Amazonian archaeology, and will be of wide interest to the readers of Nature Communications. However, I think the manuscript needs some revision and further analyses before the conclusions of the authors can be fully justified. I have described these in detail below.

The introduction sets up a major dichotomy between pre-Columbian populations in the interfluvial forests (traditionally believed to contain less dense populations) versus riverine forests (traditionally believed to hold higher population densities). Near the bottom of P3, however, the previously identified ceremonial centers in Acre, Brazil are described as being "generally far (1.5-8 km) from navigable river courses". The 1.5 – 8 km distances from navigable river courses are not "far", are within a day's walking distance of a river, and are equal to distances reported in previous literature suggesting that most pre-Columbian settlements were located within 5-10km of a major river (McMichael et al., 2012; McMichael et al., 2014; Bush et al., 2015; McMichael et al., 2017). Thus the statement by Souza et al about the ceremonial centers (which have the highest density of any of the region of earthworks) being within 1.5 – 8 km from a river supports the idea that most human settlements were close to rivers, and does not support their own conclusion statements. Souza et al. then go on to describe the previously identified fortified settlements, saying "The proximity to water sets this type of site apart from the ceremonial geometric enclosures, as they are located in small plateaus overlooking rivers". Souza et al. do not report on whether the previously identified mounded ring villages tend to be located near rivers, and I would suggest adding a sentence or two to describe that relationship. The new data that Souza et al provide on the ditched enclosures show that they "are situated on the tops of small plateaus, overlooking rivers and streams (Fig. 4a)", suggesting that this type of feature in the headwaters of the Tapajos is also related to proximity to river.

Even after Souza et al describe the relationships of these features and sites with proximity to river, the Maxent model showed that distance to river had zero importance (Supplementary Table 6), leading the authors to conclude that earthwork locations are not associated with proximity to river. I find this hard to believe with all of the descriptions given by the authors (see previous paragraph), and with looking at the maps and figures presented by the authors. Figure 5 shows that most sites are near rivers, and this figure only shows the largest of the rivers in the area. It is unclear whether these were the only rivers used as predictors in the model, or whether smaller rivers were also included. The methods section states "Distance from rivers was generated in ArcGIS using data from <http://www.hydrosheds.org/>". The HydroSheds database can be used to define rivers of any size, and the authors do not report what values they used to define their rivers. This is

crucial to support their conclusions.

The probability curve for the distance to river variable looks a bit strange, especially compared with the other predictor variables (Supplementary Figure 5), suggesting that the input raster may have had a problem. I would suggest that the authors check this predictor variable in their model, plot the distance to river raster as a supplementary figure, and report the values that they used to define rivers. If this predictor variable is key to the main conclusions, this part of the model set-up needs to be more clearly reported and defined. But even given the problems that the input raster may have had, and the lack of permutation importance in the model, earthwork probabilities still significantly decrease at distances over ca. 5 km (Supplementary Figure 5). These results do not support the main conclusions by the authors, but I think there are also several other problems with the model set up that need to be addressed (see below).

The authors report that they used the extent of the Amazon Basin for the model, but the response curves (Supplementary Figure 5) show that the potential range of elevation extends to 6000 masl, and mean annual temperature can range down to 0C. Figure 1a also shows that the defined study area exceeds that of what is considered tropical rainforest. I think it is sensible to include the savanna areas to the south of the rainforest, because the southern Amazonian rim is the focus of the manuscript. I think including the Andean regions, however, can skew the results and is the reason why elevation was deemed such an important predictor variable (this can also be seen in Supplementary Figure 5). The inclusion of the Andean regions is also why the model appears to be overfit, with AUC values > 0.96. I would suggest re-running the model only in regions that are expected to have earthworks, such as lowland Amazonian forests and the forest-savanna transition area. I do not doubt the authors' conclusions that seasonality plays a large role in driving the earthwork distributions, but I think the results (especially the importance of distance to river) may change quite a bit if the authors adjust their overall study area to fit their main questions. In general, I would suggest adding more clarity in the description of the predictor variables.

A further question on the model set-up: why were soils (e.g. FAO/IIASA/ISRIC/ISS-CAS/JRC, 2009) not included as predictor variables in the model? Soils are occasionally important predictors in predicting distributions of pre-Columbian people and their influences on forest composition (McMichael et al., 2014; Levis et al., 2017). Because the earthworks have ditches that are potentially related to drainage of some sort, the soil types may be important predictor variables for earthworks.

I am not as familiar with the methodologies used to generate population estimates, but in this section the authors describe their approach very clearly. I'm sure the approach will stir up quite a bit of discussion among archaeologists and others who study pre-Columbian Amazonia. But that is a good thing, and the authors have clearly described how they derived their estimates so that other researchers can replicate or modify the approach.

Overall, I think this is an important and well-written contribution to Amazonian archaeology regarding the distributions of pre-Columbian people across the landscape. I think the

authors have clearly justified their conclusions about pre-Columbian populations being higher than previously thought in the seasonal areas along the southern Amazonian rim. But I do not think the authors have shown any evidence that supports the conclusion that the interfluvial areas were just as densely occupied as the areas in close proximity to rivers.

If the authors want to show that the distributions of earthworks are similar in the riverine and interfluvial environments (and different than previously believed in the 95% of interfluvial forests), then they should show the frequency histograms or density distributions of earthworks as a function of distance to river, and extend those distances out to 50 or 60 km (or the maximum potential distance in their study area). Showing that earthworks are just as likely to occur at distances > 10 km river as they are at distances < 10 km of a river is the only way to support their conclusion that "population densities were higher than previously thought along the seasonal transitional forests of the interfluvial areas and minor tributaries of the southern Amazonian periphery". This would be an easy analysis to perform, and I would suggest including it as a figure in main text.

Below I have listed other minor comments:

P4: "(Gadua sp.) forest and archaeological sites in the region" – should be Guadua

In the section "Modeling earthwork distributions", it is stated that previous studies have modeled the distributions of geoglyphs and terra pretas in the past. But we have also modeled the distribution of those features with other ceramic-containing sites in Amazonia, and with paleoecological sites containing maize agriculture (McMichael et al., 2017).

best regards,
Crystal McMichael

Bibliography

Bush, M.B., McMichael, C.H., Piperno, D.R., Silman, M.R., Barlow, J.B., Peres, C.A., Power, M.J. & Palace, M.W. (2015) Anthropogenic influence on Amazonian forests in prehistory: An ecological perspective. *Journal of Biogeography*, 42, 2277-2288.

FAO/IIASA/ISRIC/ISS-CAS/JRC (2009) Harmonized World Soil Database (version 1.1). In: (ed. I. Fao), Rome, Italy and Laxenburg, Austria.

Levis, C., Costa, F.R.C., Bongers, F., Peña-Claros, M., Clement, C.R., Junqueira, A.B., Neves, E.G., Tamanaha, E.K., Figueiredo, F.O.G., Salomão, R.P., Castilho, C.V., Magnusson, W.E., Phillips, O.L., Guevara, J.E., Sabatier, D., Molino, J.-F., López, D.C., Mendoza, A.M., Pitman, N.C.A., Duque, A., Vargas, P.N., Zartman, C.E., Vasquez, R., Andrade, A., Camargo, J.L., Feldpausch, T.R., Laurance, S.G.W., Laurance, W.F., Killeen, T.J., Nascimento, H.E.M., Montero, J.C., Mostacedo, B., Amaral, I.L., Guimarães Vieira, I.C., Brien, R., Castellanos, H., Terborgh, J., Carim, M.d.J.V., Guimarães, J.R.d.S., Coelho, L.d.S., Matos, F.D.d.A., Wittmann, F., Mogollón, H.F., Damasco, G., Dávila, N., García-Villacorta, R., Coronado, E.N.H., Emilio, T., Filho, D.d.A.L., Schiatti, J., Souza, P., Targhetta, N., Comiskey, J.A., Marimon, B.S., Marimon, B.-H., Neill, D., Alonso, A., Arroyo, L., Carvalho, F.A., de Souza, F.C., Dallmeier, F., Pansonato, M.P., Duivenvoorden, J.F., Fine,

P.V.A., Stevenson, P.R., Araujo-Murakami, A., Aymard C., G.A., Baraloto, C., do Amaral, D.D., Engel, J., Henkel, T.W., Maas, P., Petronelli, P., Revilla, J.D.C., Stropp, J., Daly, D., Gribel, R., Paredes, M.R., Silveira, M., Thomas-Caesar, R., Baker, T.R., da Silva, N.F., Ferreira, L.V., Peres, C.A., Silman, M.R., Cerón, C., Valverde, F.C., Di Fiore, A., Jimenez, E.M., Mora, M.C.P., Toledo, M., Barbosa, E.M., Bonates, L.C.d.M., Arboleda, N.C., Farias, E.d.S., Fuentes, A., Guillaumet, J.-L., Jørgensen, P.M., Malhi, Y., de Andrade Miranda, I.P., Phillips, J.F., Prieto, A., Rudas, A., Ruschel, A.R., Silva, N., von Hildebrand, P., Vos, V.A., Zent, E.L., Zent, S., Cintra, B.B.L., Nascimento, M.T., Oliveira, A.A., Ramirez-Angulo, H., Ramos, J.F., Rivas, G., Schöngart, J., Sierra, R., Tirado, M., van der Heijden, G., Torre, E.V., Wang, O., Young, K.R., Baider, C., Cano, A., Farfan-Rios, W., Ferreira, C., Hoffman, B., Mendoza, C., Mesones, I., Torres-Lezama, A., Medina, M.N.U., van Andel, T.R., Villarreal, D., Zagt, R., Alexiades, M.N., Balslev, H., Garcia-Cabrera, K., Gonzales, T., Hernandez, L., Huamantupa-Chuquimaco, I., Manzatto, A.G., Milliken, W., Cuenca, W.P., Pansini, S., Pauletto, D., Arevalo, F.R., Reis, N.F.C., Sampaio, A.F., Giraldo, L.E.U., Sandoval, E.H.V., Gamarra, L.V., Vela, C.I.A. & ter Steege, H. (2017) Persistent effects of pre-Columbian plant domestication on Amazonian forest composition. *Science*, 355, 925-931.

McMichael, C., Piperno, D.R., Bush, M.B., Silman, M.R., Zimmerman, A.R., Raczka, M.F. & Lobato, L.C. (2012) Sparse pre-Columbian human habitation in western Amazonia. *Science*, 336, 1429-1431.

McMichael, C., Palace, M., Bush, M., Braswell, B., Hagen, S., Neves, E., Silman, M., Tamanaha, E. & Czarnecki, C. (2014) Predicting pre-Columbian anthropogenic soils in Amazonia. *Proceedings of the Royal Society B: Biological Sciences*, 281, 20132475.

McMichael, C.N., Matthews-Bird, F., Farfan-Rios, W. & Feeley, K.J. (2017) Ancient human disturbances may be skewing our understanding of Amazonian forests. *Proceedings of the National Academy of Sciences*, 114, 522-527.

Reviewer #2 (Remarks to the Author):

I take the central point of this paper to be that a pattern in pre-Columbian earthworks has been documented in Acre, in the Llanos de Mojos, and in the Xingu, but that the intermediate region has not yet been carefully surveyed. This project examined satellite imagery, ground-truthed those images, and conducted excavations at a few of these earthworks. I am in agreement with the authors that these are a significant results, and that the distribution across 1800 kilometers is a significant fact about the archaeological record in the Southern Amazon. The pattern certainly argues for a larger interfluvial population in the late pre-Columbian period, and has implications for the rest of the Amazon, as well as interpretations of the contemporary Amazon.

The reference to Arawak speakers as an ethnographic analogy is perfectly reasonable. In light of the complicated linguistic situation across the southern Amazon, it would be good for the authors to explicitly acknowledge the multilingual nature of Amazonian peoples, especially in the Xingu and the Guaporé-Mamoré region. There are many non-Arawak

speakers represented within this large area as well as Arawak speakers.

In the classification of "Earth-builders of the southern rim of the Amazon," the category descriptions should allow for more variability, in light of the small amount of fieldwork that has been carried out, in only a few places. For example, the question about whether ditches in Mojos are fortifications or not needs to be considered on a case by case basis, not for the entire region at once. Some mention needs to be made of other earthworks as well, at least in Mojos, where circular (or roughly circular) earthworks are in some cases accompanied by raised fields, causeways and canals.

I have concerns about some of the larger conclusions that are drawn from the specifics of the new data to the entire region. The authors interpret the pattern of earthworks (i.e. Figure 5) by grouping them into three categories based on enclosure area. Why are three classes used and not four? Or two? A histogram of enclosure area would make these decisions more transparent. Does the distribution of smaller enclosures bear on the interpretation of the overall pattern? They seem to be present near some of the larger enclosures but not others. Finally, the different enclosures have not all been dated (as I understand the manuscript). How do these issues bear on the interpretation of a "peer polity"? In this short article format it is also difficult to outline the assumptions and implications of the peer-polity idea, and they are not outlined in the manuscript as written.

In "Discussions and implications," a form of peer-polity idea is extended across the entire southern Amazon, and in light of the limited amount of data available across the different areas, this conclusion might be expressed in more moderate language. Without independent evidence about political organization (from sources other than spatial patterning) more room should be kept open for different interpretations of social and political organization.

Some of the difficulties with modeling over such a large area with input from many different researches is the compatibility of different definitions of "site." In many cases, it may be hard to distinguish one set of earthworks from one another, over very long distances, and large areas. Should an extension of earthworks over several kilometers be counted as one "site" or several? Such issues would change the results of the spatial model, by changing the inputs.

There are a few slight grammatical or word use issues that are not critical, but could help the main point of the paper be delivered more effectively. In the abstract, for example, the second sentence ends with "...a large spatial gap in Amazonian archaeology," which would make more sense (I think) as "...a large spatial gap in the archaeological record."

In summary, the paper presents important new data and it should be published after revisions. The interpretation of political organization from settlement pattern should be more explicitly discussed, and not uncritically applied across all of the example regions.

John H. Walker
University of Central Florida

Reviewer #3 (Remarks to the Author):

The paper is interesting and technically sound. I have no objections to its publication. There is clearly more to learn about this exciting area of archaeological research and this is a step in the right direction.

The authors do a good job of using the summed radio-carbon curves to indicate relative increases in construction by site-type while avoiding the over-interpretation of such summed curves.

Response to reviewers

As a comment for all reviewers, we would like to point out that, since the original submission of the paper, new earthworks have been found in satellite imagery in the UTB, all of which have been incorporated into the new model.

The original reports of the reviewers are presented below in blue, and our response follows in black.

Reviewer #1

Among other comments, this reviewer raised several important questions about the predictive model and made suggestions to improve it, all of which have been incorporated.

Souza et al. present new data on 86 earthworks from 67 sites identified along the southern Amazonian rim in the headwaters of the Tapajos River, model their new data alongside the locations of previously reported earthworks, and “conclude that population densities were higher than previously thought along the seasonal transitional forests of the interfluvial areas and minor tributaries of the southern Amazonian periphery”. These data are a substantial contribution to Amazonian archaeology, and will be of wide interest to the readers of Nature Communications. However, I think the manuscript needs some revision and further analyses before the conclusions of the authors can be fully justified. I have described these in detail below.

The introduction sets up a major dichotomy between pre-Columbian populations in the interfluvial forests (traditionally believed to contain less dense populations) versus riverine forests (traditionally believed to hold higher population densities). Near the bottom of P3, however, the previously identified ceremonial centers in Acre, Brazil are described as being “generally far (1.5-8 km) from navigable river courses”. The 1.5 – 8 km distances from navigable river courses are not “far”, are within a day’s walking distance of a river, and are equal to distances reported in previous literature suggesting that most pre-Columbian settlements were located within 5-10km of a major river (McMichael et al., 2012; McMichael et al., 2014; Bush et al., 2015; McMichael et al., 2017). Thus the statement by Souza et al about the ceremonial centers (which have the highest density of any of the region of earthworks) being within 1.5 – 8 km from a river supports the idea that most human settlements were close to rivers, and does not support their own conclusion statements. Souza et al. then go on to describe the previously identified fortified settlements, saying “The proximity to water sets this type of site apart from the ceremonial geometric enclosures, as they are located in small plateaus overlooking rivers”. Souza et al. do not report on whether the previously identified mounded ring villages tend to be located near rivers, and I would suggest adding a sentence or two to describe that relationship. The new data that Souza et al provide on the ditched enclosures show that they “are situated on the tops of small plateaus, overlooking rivers and streams (Fig. 4a)”, suggesting that this type of feature in the headwaters of the Tapajos is also related to proximity to river.

The distance of geoglyphs from water is no longer described as “far”, which was meant to be understood in relative terms (e.g. compared to ADE sites, see below) (p. 6, l. 2-3). As suggested, we describe the placement of mounded villages in the landscape (p. 8, l. 2-4).

Even after Souza et al describe the relationships of these features and sites with proximity to river, the Maxent model showed that distance to river had zero importance (Supplementary Table 6), leading the authors to conclude that earthwork locations are not associated with proximity to river. I find this hard to believe with all of the descriptions given by the authors (see previous paragraph), and with looking at the maps and figures presented by the authors. Figure 5 shows that most sites

are near rivers, and this figure only shows the largest of the rivers in the area. It is unclear whether these were the only rivers used as predictors in the model, or whether smaller rivers were also included. The methods section states “Distance from rivers was generated in ArcGIS using data from <http://www.hydrosheds.org/>”. The HydroSheds database can be used to define rivers of any size, and the authors do not report what values they used to define their rivers. This is crucial to support their conclusions.

The probability curve for the distance to river variable looks a bit strange, especially compared with the other predictor variables (Supplementary Figure 5), suggesting that the input raster may have had a problem. I would suggest that the authors check this predictor variable in their model, plot the distance to river raster as a supplementary figure, and report the values that they used to define rivers. If this predictor variable is key to the main conclusions, this part of the model set-up needs to be more clearly reported and defined. But even given the problems that the input raster may have had, and the lack of permutation importance in the model, earthwork probabilities still significantly decrease at distances over ca. 5 km (Supplementary Figure 5). These results do not support the main conclusions by the authors, but I think there are also several other problems with the model set up that need to be addressed (see below).

The flow accumulation threshold used to define navigable rivers is now explicitly reported (p. 17, l. 1-2). We used a value of 15,000 in line with previous works, allowing us to compare the results (e.g. McMichael et al. 2014, see below). A new distance raster was generated and the model rerun, since the reviewer raised concerns that the original raster might have had a problem.

The authors report that they used the extent of the Amazon Basin for the model, but the response curves (Supplementary Figure 5) show that the potential range of elevation extends to 6000 masl, and mean annual temperature can range down to 0C. Figure 1a also shows that the defined study area exceeds that of what is considered tropical rainforest. I think it is sensible to include the savanna areas to the south of the rainforest, because the southern Amazonian rim is the focus of the manuscript. I think including the Andean regions, however, can skew the results and is the reason why elevation was deemed such an important predictor variable (this can also be seen in Supplementary Figure 5). The inclusion of the Andean regions is also why the model appears to be overfit, with AUC values > 0.96. I would suggest re-running the model only in regions that are expected to have earthworks, such as lowland Amazonian forests and the forest-savanna transition area. I do not doubt the authors’ conclusions that seasonality plays a large role in driving the earthwork distributions, but I think the results (especially the importance of distance to river) may change quite a bit if the authors adjust their overall study area to fit their main questions. In general, I would suggest adding more clarity in the description of the predictor variables.

The new model was also run using a smaller area, because the reviewer made the important point that the inclusion of Andean regions might have skewed the results. Therefore, we clipped all the rasters to the extent of the Amazon basin excluding elevations > 1000 m, while maintaining the transitional forests and savannas south of the basin which are crucial for the distribution of the earthworks analysed (p. 18, l. 2-5).

A further question on the model set-up: why were soils (e.g. FAO/IIASA/ISRIC/ISS-CAS/JRC, 2009) not included as predictor variables in the model? Soils are occasionally important predictors in predicting distributions of pre-Columbian people and their influences on forest composition

(McMichael et al., 2014; Levis et al., 2017). Because the earthworks have ditches that are potentially related to drainage of some sort, the soil types may be important predictor variables for earthworks. As suggested by the reviewer, we ran the new model including soil predictors derived from the harmonized world soil database (p. 17, l. 26-27 and p. 18, l. 14-16; Supplementary Tables 5-6; Supplementary Figs. 6 and 9).

I am not as familiar with the methodologies used to generate population estimates, but in this section the authors describe their approach very clearly. I'm sure the approach will stir up quite a bit of discussion among archaeologists and others who study pre-Columbian Amazonia. But that is a good thing, and the authors have clearly described how they derived their estimates so that other researchers can replicate or modify the approach.

Overall, I think this is an important and well-written contribution to Amazonian archaeology regarding the distributions of pre-Columbian people across the landscape. I think the authors have clearly justified their conclusions about pre-Columbian populations being higher than previously thought in the seasonal areas along the southern Amazonian rim. But I do not think the authors have shown any evidence that supports the conclusion that the interfluvial areas were just as densely occupied as the areas in close proximity to rivers.

If the authors want to show that the distributions of earthworks are similar in the riverine and interfluvial environments (and different than previously believed in the 95% of interfluvial forests), then they should show the frequency histograms or density distributions of earthworks as a function of distance to river, and extend those distances out to 50 or 60 km (or the maximum potential distance in their study area). Showing that earthworks are just as likely to occur at distances > 10 km river as they are at distances < 10 km of a river is the only way to support their conclusion that "population densities were higher than previously thought along the seasonal transitional forests of the interfluvial areas and minor tributaries of the southern Amazonian periphery". This would be an easy analysis to perform, and I would suggest including it as a figure in main text.

All the changes suggested by the reviewer helped to improve the model, but the results remained essentially similar (Figure 1, Supplementary Fig. 11), with bioclimatic variables still the most important predictors for earthwork distribution. Distance to rivers indeed gained some importance in relation to the previous model (p. 19, l. 9-11; Supplementary Tables 5-6), but it is certainly not as relevant a predictor as it is for ADE sites.

With all that in mind, we added a more explicit discussion comparing our modelled distribution of geometrical earthworks with that of ADE sites reported in McMichael et al. 2014 (p. 14, l. 11-22). As discussed in that paragraph, we are confident that these two types of sites conform to very distinct settlement patterns.

Finally, this reviewer suggested that site frequency should be plotted as a function of distance to rivers in order to support our conclusion that interfluvial areas of the SRA were more densely settled than previously thought. We have performed that analysis, both for all the SRA and for its specific sub-regions, showing that variations exist (Figure 6, p. 14, l. 17-22). As is clear from the results, however, we believe that the condition of "showing that earthworks are just as likely to occur at distances > 10 km river as they are at distances < 10 km of a river" has been fulfilled.

Below I have listed other minor comments:

P4: "(*Gadua* sp.) forest and archaeological sites in the region" – should be *Guadua*

In the section "Modeling earthwork distributions", it is stated that previous studies have modeled the distributions of geoglyphs and terra pretas in the past. But we have also modeled the distribution of those features with other ceramic-containing sites in Amazonia, and with paleoecological sites containing maize agriculture (McMichael et al., 2017).

The minor corrections (in the spelling of *Guadua* and in the acknowledgement of more previous modelling studies) have been made (p. 6, l. 4 and p. 13, l. 14-17).

As a final comment to Reviewer #1, we would like to stress that we do not doubt that major navigable rivers were important foci of Pre-Columbian populations (clear from the distribution of ADE sites). However, our main point is that substantial areas further from the larger floodplains were until recently thought to be sparsely occupied and with negligible impact on the landscape (e.g. the UTB as it appears in Levis et al. 2017), while our evidence shows that population densities on some of these areas were higher than previously thought (p. 2, l. 3-5; p. 16, l. 10-14).

Reviewer #2

This reviewer questioned some of our broader claims regarding the political organisation of Pre-Columbian populations in the SRA, among other suggestions.

I take the central point of this paper to be that a pattern in pre-Columbian earthworks has been documented in Acre, in the Llanos de Mojos, and in the Xingu, but that the intermediate region has not yet been carefully surveyed. This project examined satellite imagery, ground-truthed those images, and conducted excavations at a few of these earthworks. I am in agreement with the authors that these are a significant results, and that the distribution across 1800 kilometers is a significant fact about the archaeological record in the Southern Amazon. The pattern certainly argues for a larger interfluvial population in the late pre-Columbian period, and has implications for the rest of the Amazon, as well as interpretations of the contemporary Amazon.

The reference to Arawak speakers as an ethnographic analogy is perfectly reasonable. In light of the complicated linguistic situation across the southern Amazon, it would be good for the authors to explicitly acknowledge the multilingual nature of Amazonian peoples, especially in the Xingu and the Guaporé-Mamoré region. There are many non-Arawak speakers represented within this large area as well as Arawak speakers.

We have acknowledged the high linguistic diversity of southern Amazonia and the multi-ethnic/multilinguistic nature of some regional systems (p. 4, l. 12-15).

In the classification of "Earth-builders of the southern rim of the Amazon," the category descriptions should allow for more variability, in light of the small amount of fieldwork that has been carried out, in only a few places. For example, the question about whether ditches in Mojos are fortifications or not needs to be considered on a case by case basis, not for the entire region at once. Some mention needs to be made of other earthworks as well, at least in Mojos, where circular (or roughly circular) earthworks are in some cases accompanied by raised fields, causeways and canals.

We have also mentioned other potential functions of ditched enclosures (p. 7, l. 1-4). The reviewer also suggests that we mention other types of earthworks in the Llanos de Moxos (raised fields, canals). This was already present in the original manuscript (now in p. 3, l. 24-27).

I have concerns about some of the larger conclusions that are drawn from the specifics of the new data to the entire region. The authors interpret the pattern of earthworks (i.e. Figure 5) by grouping them into three categories based on enclosure area. Why are three classes used and not four? Or two? A histogram of enclosure area would make these decisions more transparent. Does the distribution of smaller enclosures bear on the interpretation of the overall pattern? They seem to be present near some of the larger enclosures but not others.

This reviewer suggested that we should present histograms of site area in order to support classification into size categories. We have included that information (Supplementary Fig. 1) and revised our categorization. We have explored the consequences of using different categorizations of site size for settlement patterns (p. 17, l. 9-19; Supplementary Fig. 2). We confirm that small sites are significantly clustered, while the largest sites are regularly spaced.

Finally, the different enclosures have not all been dated (as I understand the manuscript). How do these issues bear on the interpretation of a “peer polity”? In this short article format it is also difficult to outline the assumptions and implications of the peer-polity idea, and they are not outlined in the manuscript as written.

We have now discussed in more detail our understanding of a peer-polity system as recorded in the ethnography of the Upper Xingu and its projection to the archaeology of that region, with which we find major similarities in the settlement patterns of the UTB (p. 11, l. 17-27 and p. 12, l. 1-4).

In “Discussions and implications,” a form of peer-polity idea is extended across the entire southern Amazon, and in light of the limited amount of data available across the different areas, this conclusion might be expressed in more moderate language. Without independent evidence about political organization (from sources other than spatial patterning) more room should be kept open for different interpretations of social and political organization.

We have used a more moderate language in interpreting the political organization of Pre-Columbian societies along the SRA. The fact that hierarchical settlement patterns have been found in our area and in the Upper Xingu, but not elsewhere has been highlighted (p. 12, l. 5-11) and we stress these variations in other parts of the manuscript (e.g. p. 2, l. 10-11).

Some of the difficulties with modeling over such a large area with input from many different researches is the compatibility of different definitions of “site.” In many cases, it may be hard to distinguish one set of earthworks from one another, over very long distances, and large areas. Should an extension of earthworks over several kilometers be counted as one “site” or several? Such issues would change the results of the spatial model, by changing the inputs.

This reviewer raised a very important point about the definition of archaeological sites, which is somewhat arbitrary in the case of earthworks found in close proximity. Because this could indeed skew the results of the predictive model, we tested, beyond the original dataset, filtered datasets where sites occurring within 2.5 and 5 km of each other were considered a single site (p. 18, l. 25-30). Although the results were not different, we used the 2.5 km threshold as a compromise.

There are a few slight grammatical or word use issues that are not critical, but could help the main point of the paper be delivered more effectively. In the abstract, for example, the second sentence

ends with "...a large spatial gap in Amazonian archaeology," which would make more sense (I think) as "...a large spatial gap in the archaeological record."

The grammatical correction in the abstract has been made (p. 2, l. 7).

In summary, the paper presents important new data and it should be published after revisions. The interpretation of political organization from settlement pattern should be more explicitly discussed, and not uncritically applied across all of the example regions.

Reviewer #3

This reviewer did not suggest any changes.

The paper is interesting and technically sound. I have no objections to its publication. There is clearly more to learn about this exciting area of archaeological research and this is a step in the right direction.

The authors do a good job of using the summed radio-carbon curves to indicate relative increases in construction by site-type while avoiding the over-interpretation of such summed curves.

REVIEWERS' COMMENTS:

Reviewer #1 (Remarks to the Author):

As I stated in the original review, I think this manuscript provides a very important contribution to the discussion on the distributions of pre-Columbian people in Amazonia. I think the revisions that you have incorporated definitely make the manuscript stronger, although as a whole, I think the manuscript still needs a bit of restructuring. I think the results, interpretation, and take-home message would be much clearer and stronger if the text was restructured a bit. As it is, there are results interspersed with methods, which lead to some very important points being overlooked and buried within the methods section. I have provided more detailed suggestions on how this can be made clearer in the attached pdf document. I hope that you find the reviews and comments helpful in improving the clarity of the manuscript.

Reviewer #2 (Remarks to the Author):

I appreciate the authors' hard work both in the research and writing of the first version of the article, and in the revisions which they undertook in response to all the reviewers' comments. I have read and examined the changes that were made and I feel that the points I raised have been satisfactorily addressed.

I would also suggest that the viewpoints represented by the authors and the three reviewers displays a healthy diversity of informed opinion within Amazonian studies.

John H. Walker
University of Central Florida

Summary of Comments from Reviewer #1

Page: 2

 Author: cmcmicha Subject: Sticky Note Date: 1/24/2018 5:59:11 PM
I would suggest re-wording this sentence...it reads a bit awkwardly

 Author: cmcmicha Subject: Sticky Note Date: 1/24/2018 6:00:05 PM
I think using this word is quite strong, and should be adjusted according to the results.

Page: 5

 Author: cmcmicha Subject: Sticky Note Date: 1/24/2018 6:12:16 PM
an 1800-km transect? or a square area? can you clarify and refer to a figure? is this the UTB red box on fig 1a? The red box is not labeled in the caption for Figure 1.

 Author: cmcmicha Subject: Sticky Note Date: 1/26/2018 11:33:26 AM
this classification system is more of a 'site description' and 'methods' sections. I would suggest moving it accordingly. I think you've stated your objectives above and should move into the main results here (many of which i have highlighted in yellow below). The methods section should come at the end, which is standard for publications in Nature journals.

 Author: cmcmicha Subject: Sticky Note Date: 1/26/2018 11:32:40 AM
i think this is unnecessary text, and this point has been raised in other places in the manuscript

 Author: cmcmicha Subject: Cross-Out Date: 1/26/2018 11:32:13 AM

 Author: cmcmicha Subject: Sticky Note Date: 1/26/2018 11:37:05 AM
this sentence is difficult to read. i would suggest re-wording

 Author: cmcmicha Subject: Highlight Date: 1/26/2018 11:36:43 AM

Page: 6

 Author: cmcmicha Subject: Sticky Note Date: 1/26/2018 2:12:12 PM
these are very important results whereas the above text is more site description/methods. I would suggest restructuring much of these sections and below to make clear what is background and methods and what are the important results from your analyses

 Author: cmcmicha Subject: Highlight Date: 1/26/2018 2:11:11 PM

 Author: cmcmicha Subject: Sticky Note Date: 1/26/2018 2:14:37 PM
This statement goes against some of the statements in the abstract and conclusions, which say that distance to river was relatively unimportant in determining the overall 'earthwork' distribution.

 Author: cmcmicha Subject: Highlight Date: 1/26/2018 2:13:19 PM

Page: 7

 Author: cmcmicha Subject: Sticky Note Date: 1/26/2018 2:17:03 PM
be specific here - describe which new data support this idea and have it placed solidly with all other results

Page: 8

T Author: cmcmicha Subject: Highlight Date: 1/26/2018 2:20:06 PM
same comment as above in regards to this landscape configuration and the conclusions presented in the manuscript

☞ Author: cmcmicha Subject: Sticky Note Date: 1/26/2018 2:22:37 PM
Besides the text I highlighted in the above sections (important results) the above sections read like site descriptions/
background information.

The survey below reads as methods - I would strongly suggest restructuring in a more traditional format of Nature journals, where the results and discussion follow the introduction, and the site descriptions/methods are presented at the end of the manuscript or in supplementary text (depending on length)

☞ Author: cmcmicha Subject: Sticky Note Date: 1/26/2018 2:23:04 PM
archaeological survey or satellite imagery survey?

☞ Author: cmcmicha Subject: Sticky Note Date: 1/26/2018 2:24:16 PM
this should be shown in figure 1, and somewhere in the methods you should clarify how these two study sites (this one and the 1800km² one) have been surveyed...is one through archaeological excavations and the other with remote sensing? or do both areas have both types of surveys?

☞ Author: cmcmicha Subject: Sticky Note Date: 1/26/2018 2:25:13 PM
here it leads me to think this is written about previous archaeological excavations??

this paragraph/section is a bit unclear and needs some clarification

Page: 9

☞ Author: cmcmicha Subject: Sticky Note Date: 1/26/2018 2:25:37 PM
needs references

☞ Author: cmcmicha Subject: Sticky Note Date: 1/26/2018 2:25:57 PM
this and previous sentence need references

☞ Author: cmcmicha Subject: Sticky Note Date: 1/26/2018 2:26:59 PM
are these results from your archaeological surveys performed in this manuscript or from previous research? it is a bit unclear here

☞ Author: cmcmicha Subject: Sticky Note Date: 1/26/2018 2:27:43 PM
in the 54,000 km² area? or in the 1800 km² area?

☞ Author: cmcmicha Subject: Sticky Note Date: 1/26/2018 2:30:20 PM
are these sites that were used to build your predictive model or test it?

☞ Author: cmcmicha Subject: Sticky Note Date: 1/26/2018 2:31:24 PM
this is really methods, correct? I had suggested in your methods section to clarify how the classification was performed. I would suggest moving this to the other section

T Author: cmcmicha Subject: Highlight Date: 1/26/2018 2:30:46 PM

Page: 10

Author: cmcmicha Subject: Sticky Note Date: 1/26/2018 2:54:52 PM

I think it is worth adding some text to hypothesize why this site is so much larger than the rest in the discussion. Is this site special in its location? soil types? etc?

Author: cmcmicha Subject: Sticky Note Date: 1/26/2018 2:55:41 PM

the positioning of this site is hard to see in fig 4a.

Author: cmcmicha Subject: Sticky Note Date: 1/26/2018 2:56:15 PM

intensity being a sedentary society here?

Author: cmcmicha Subject: Sticky Note Date: 1/26/2018 2:56:41 PM

can this be shown on a figure?

Author: cmcmicha Subject: Sticky Note Date: 1/26/2018 2:57:24 PM

i would suggest a figure of only the study area where you can show the distributions etc of these different types of features

Page: 11

Author: cmcmicha Subject: Sticky Note Date: 1/26/2018 2:59:25 PM

these results are all occurring inside the red box shown in fig 1a? this is unclear where the new results are coming from.

again, i would suggest a figure showing a close-up view of your overall study area and the distributions of these features within that study area

Author: cmcmicha Subject: Sticky Note Date: 1/26/2018 3:01:33 PM

I would suggest drawing boxes around the areas of interest in fig 4e instead of drawing on top of the features, which make it impossible to see the actual features

Author: cmcmicha Subject: Sticky Note Date: 1/26/2018 3:02:35 PM

these are actually important results and should be clearly placed in a results section

Author: cmcmicha Subject: Highlight Date: 1/26/2018 3:02:13 PM

Author: cmcmicha Subject: Sticky Note Date: 1/26/2018 3:05:47 PM

same comment as previous - this is really the crux of the results - and it gets diluted in the text that contains so much site description and methods

Author: cmcmicha Subject: Highlight Date: 1/26/2018 3:05:05 PM

Page: 12

Author: cmcmicha Subject: Highlight Date: 1/26/2018 3:18:07 PM

this section again jumps back to site descriptions (based on previous work) and methods for the current analysis, but is interspersed with results

I would suggest restructuring into clearly defined results/discussion and methods sections.

Page: 13

Author: cmcmicha Subject: Sticky Note Date: 1/26/2018 3:19:10 PM

it is unclear whether the model was used to predict sites that you tested with excavations or whether the excavations were sites included in the model

Page: 14

Author: cmcmicha Subject: Sticky Note Date: 1/26/2018 3:24:55 PM
two periods here - one before and one after the reference. please remove one of them

Author: cmcmicha Subject: Sticky Note Date: 1/26/2018 3:25:43 PM
can you report specific distances in comparison with the ADE results?

Author: cmcmicha Subject: Sticky Note Date: 1/26/2018 3:26:32 PM
this is very important - and I think it would strengthen the manuscript to provide the exact numbers for the results between regions and site types

Author: cmcmicha Subject: Inserted Text Date: 1/26/2018 3:29:58 PM
s

Page: 15

Author: cmcmicha Subject: Highlight Date: 1/26/2018 3:32:48 PM
are the exact density numbers reported in your results section? that is also an important result

Author: cmcmicha Subject: Highlight Date: 1/26/2018 3:36:31 PM
is this based on environmental similarity across the region or purely on geographic area? I think this should be clarified, especially given the following sentences

Page: 16

Author: cmcmicha Subject: Sticky Note Date: 1/26/2018 4:59:13 PM
uninterrupted area sounds like they are continuous across the landscape...maybe for clarification it would be better to report the density and geographic spread along with this statement

Author: cmcmicha Subject: Sticky Note Date: 1/26/2018 5:00:22 PM
I would suggest defining the sub-regions or maybe showing them on a figure

Page: 17

Author: cmcmicha Subject: Sticky Note Date: 1/26/2018 10:44:11 AM
is this the red box in figure 1a? please clarify

Author: cmcmicha Subject: Sticky Note Date: 1/26/2018 10:45:23 AM
It is still a bit unclear in the manuscript whether this imagery was used to test the maxent model predictions, or whether it was used to provide occurrence locations to input into the model in the data-sparse region

Author: cmcmicha Subject: Sticky Note Date: 1/26/2018 10:50:15 AM
can you define size classifications?

Author: cmcmicha Subject: Highlight Date: 1/26/2018 10:51:58 AM
this is actually part of your results, and is quite interesting. i would suggest moving this text into your results section

Page: 18

-
- Author: cmcmicha Subject: Sticky Note Date: 1/26/2018 10:54:41 AM
Food for thought here...it would be nice to have constructed the model using only the previously published occurrence locations from both the eastern and western sections. Then test the model using your imagery and newly detected sites. It would be a really cool way to show how the model performs in underrepresented locations, and an assessment of its robustness in certain conditions. If you don't want to reconfigure this manuscript, then I would suggest that we team up in the future and try an approach like this on a variety of site types and across a variety of regions.
-
- Author: cmcmicha Subject: Sticky Note Date: 1/26/2018 11:00:01 AM
Here I think it would be good to define the area included in your study area - figure 1 looks to be defined as Amazonia sensu stricto (according to Eva et al. 2005) and defined as all regions draining into the Amazon River system and that are less than 1000 m elevation
-
- Author: cmcmicha Subject: Inserted Text Date: 1/26/2018 10:57:32 AM
(Fig. 1)
-
- Author: cmcmicha Subject: Sticky Note Date: 1/26/2018 11:00:29 AM
what were the criteria for eliminating predictors? please define
-
- Author: cmcmicha Subject: Sticky Note Date: 1/26/2018 11:02:25 AM
in a manuscript that i currently have in review, i am getting criticized for using some of these variables - particularly ones that are hard to interpret ecologically. i agree with the reviewers, and will construct future models with a reduced set of predictor variables. maybe something to think about here as well?
-
- Author: cmcmicha Subject: Sticky Note Date: 1/26/2018 11:03:01 AM
insert references here

Page: 19

-
- Author: cmcmicha Subject: Sticky Note Date: 1/26/2018 11:05:06 AM
should kilometer be spelled out here?
-
- Author: cmcmicha Subject: Sticky Note Date: 1/26/2018 11:06:50 AM
are these results shown in the supplementary material? i would suggest showing the data, and citing the appropriate figure/table here
-
- Author: cmcmicha Subject: Inserted Text Date: 1/26/2018 11:07:02 AM
tests
-
- Author: cmcmicha Subject: Sticky Note Date: 1/26/2018 11:08:46 AM
this is all results, and should be moved to the results section of the main text or of the supplementary material
-
- Author: cmcmicha Subject: Highlight Date: 1/26/2018 11:11:09 AM
incomplete comparison here - in relation to which other variables? all of them? if it is the second-most important overall, then I think some of the text and interpretation should be revisited.
-
- Author: cmcmicha Subject: Inserted Text Date: 1/26/2018 11:07:26 AM
the importance of the predictor variables
-
- Author: cmcmicha Subject: Sticky Note Date: 1/26/2018 11:13:04 AM
This bit of text is confusing. It would be better to have a table of those variable importances for the reader to refer to - like supplementary table 6.
-
- Author: cmcmicha Subject: Inserted Text Date: 1/26/2018 11:07:43 AM
a
-
- Author: cmcmicha Subject: Sticky Note Date: 1/26/2018 11:10:21 AM
how does it compare with the bioclim predictors?
-
- Author: cmcmicha Subject: Inserted Text Date: 1/26/2018 11:13:41 AM

Page: 20

Author: cmcmicha Subject: Sticky Note Date: 1/26/2018 11:14:27 AM
not needed since it was said in the previous sentence

Author: cmcmicha Subject: Sticky Note Date: 1/26/2018 11:17:01 AM
this is also results, and i would do a bit of restructuring in this section

Author: cmcmicha Subject: Cross-Out Date: 1/26/2018 11:14:12 AM

Author: cmcmicha Subject: Highlight Date: 1/26/2018 11:16:37 AM

Author: cmcmicha Subject: Sticky Note Date: 1/26/2018 11:18:08 AM
this is back to methods. I think the manuscript would be greatly improved by some restructuring of the methods section, which currently includes a substantial amount of results that are important to the main message of the paper

Author: cmcmicha Subject: Inserted Text Date: 1/26/2018 11:19:06 AM
t

Author: cmcmicha Subject: Sticky Note Date: 1/26/2018 11:21:03 AM
these are results and discussion that are central to the main message of the paper. I think these results should be presented outside of the methods section

Author: cmcmicha Subject: Highlight Date: 1/26/2018 11:20:22 AM

Page: 21

Author: cmcmicha Subject: Inserted Text Date: 1/24/2018 6:03:24 PM
S

Author: cmcmicha Subject: Sticky Note Date: 1/24/2018 6:04:26 PM
DOI is included for some, but not all, references. keep consistent in formatting

Page: 28

Author: cmcmicha Subject: Sticky Note Date: 1/26/2018 10:43:14 AM
clarify whether these are sites reported in previous literature or those being reported in this study.

define red box (study area of 1800km2??)

provide references for "other sites" in the caption"

Author: cmcmicha Subject: Inserted Text Date: 1/26/2018 2:03:36 PM
showing

In what follows, we reply to the comments of Reviewer #1. The suggestions made by that reviewer in the annotated pdf file are presented in blue, and our response follows in black.

Response to comments of Reviewer #1

p. 2 (“... major floodplains.”) I would suggest re-wording this sentence...it reads a bit awkwardly

This sentence has been re-worded, together with other sentences in the abstract in order to conform to 150 words.

p. 2 (referring to “entirety”) I think using this word is quite strong, and should be adjusted according to the results.

The word has been removed.

p. 5 an 1800-km transect? or a square area? can you clarify and refer to a figure? is this the UTB red box on fig 1a? The red box is not labeled in the caption for Figure 1.

The distance has been clarified as referring to an East-West transect. Reference to Figure 1 has been added accordingly. We have added the description of the red box and UTB as Upper Tapajós Basin to the caption of Figure 1, and referred to Figure 4 as showing that region in detail.

p. 5 (“Earth-builders of the southern...”) this classification system is more of a 'site description' and 'methods' sections. I would suggest moving it accordingly. I think you've stated your objectives above and should move into the main results here (many of which i have highlighted in yellow below). The methods section should come at the end, which is standard for publications in Nature journals.

We have moved this section to Supplementary Text.

p. 5 (“Beyond their general...”) i think this is unnecessary text, and this point has been raised in other places in the manuscript

This sentence has been removed.

p. 5 (“Large-scale deforestation...”) this sentence is difficult to read. i would suggest re-wording

We have re-written the sentence.

p. 6 (“The closer proximity...”) This statement goes against some of the statements in the abstract and conclusions, which say that distance to river was relatively unimportant in

determining the overall 'earthwork' distribution. and p. 8 ("Thus, their landscape...") same comment as above in regards to this landscape configuration and the conclusions presented in the manuscript

This type of observation has been repeatedly made by Reviewer #1. However, **it is impossible to find anywhere in the Amazon that is "far" from any source of water**. The main question is how to define a "navigable" or "perennial" river. In our model, we use exactly the same criteria used by Reviewer #1 in her published work (>15,000 flow accumulation). Under that threshold, proximity to rivers turned out to be an insignificant predictor for ditched enclosures, unlike her own model of ADE sites using the same criteria. Thus, even though many earthworks are close to springs or small creeks, they are far from navigable/perennial rivers. We believe this is very clearly explained in the results and methods section.

p. 6 ("A compilation of the available...") these are very important results whereas the above text is more site description/methods. I would suggest restructuring much of these sections and below to make clear what is background and methods and what are the important results from your analyses

This text and others referring to the observed trends in the radiocarbon dates have been moved to the results section. The remaining descriptive text has been moved to the methods section.

p. 7 ("The habitation nature...") be specific here - describe which new data support this idea and have it placed solidly with all other results

We have described the presence of ADE and high density of ceramics in the UTB sites. We have added this as supporting the habitation nature of the enclosures in the results section, when describing the excavations at Mt-04.

p. 8 ("Survey in the Upper...") Besides the text I highlighted in the above sections (important results) the above sections read like site descriptions/background information.

The survey below reads as methods - I would strongly suggest restructuring in a more traditional format of Nature journals, where the results and discussion follow the introduction, and the site descriptions/methods are presented at the end of the manuscript or in supplementary text (depending on length)

We have re-structured the manuscript accordingly. Methods of the survey were moved to the methods section, while we present the general characterisation of the sites and settlement patterns of the UTB in the results section.

p. 8 archaeological survey or satellite imagery survey?

We have explained that we conducted satellite imagery survey, and later explain that our research was multi-staged, with fieldwork directed to check a selection of archaeological sites identified through satellite imagery.

p. 8 (“... comprising the basins ...”) this should be shown in figure 1, and somewhere in the methods you should clarify how these two study sites (this one and the 1800km² one) have been surveyed...is one through archaeological excavations and the other with remote sensing? or do both areas have both types of surveys?

The surveyed area has now been labelled in the caption of Fig. 1a, and we refer the reader to this figure and to Fig. 5 where the surveyed area with the named rivers and all sites is shown in detail. We also refer to previous works combining satellite imagery and ground survey in other parts of the SRA.

p. 8 (“Those projects...”) here it leads me to think this is written about previous archaeological excavations??

this paragraph/section is a bit unclear and needs some clarification

We removed the text referring to research in the lower Tapajos, as we judged it overall irrelevant for the argument of the paper.

p. 9 (“... Columbian encounter.”) needs references

References have been added.

p. 9 (“... have been recorded.”) this and previous sentence need references

References have been added.

p. 9 (“Archaeological sites are restricted...”) are these results from your archaeological surveys performed in this manuscript or from previous research? it is a bit unclear here

We have moved these sentences, which were a bit misplaced, to the following paragraph, where it is clear that we are referring to the results of our survey.

p. 9 (“We designed...”) in the 54,000 km² area? or in the 1800 km² area?

We now explain that this refers to the selected area of the UTB.

p. 9 (“... 81 potential pre-Columbian...”) are these sites that were used to build your predictive model or test it?

The methods section explains that the sites used for building the predictive model were all previously recorded sites + our new sites. We also explain how each of the 10 iterations of the model used a random 25% testing sample from all sites (including the new ones).

p. 9 (“We propose a categorisation...”) this is really methods, correct? I had suggested in your methods section to clarify how the classification was performed. I would suggest moving this to the other section

The method and values used in the classification are now in the Methods section, but the general categories of sites based on those values are reported in the Results section.

p. 10 (“... area of ~19 ha”) I think it is worth adding some text to hypothesize why this site is so much larger than the rest in the discussion. Is this site special in its location? soil types? etc?

At this moment, we prefer not to speculate on why this site is so large and why it is located where it is, but these are certainly questions that we have in mind for future research.

p. 10 (“Fig4a”) the positioning of this site is hard to see in fig 4a.

The authors disagree. We think the photo shows the enclosure and its position on the landscape quite clearly.

p. 10 (“... and tend to appear isolated.”) i would suggest a figure of only the study area where you can show the distributions etc of these different types of features

Figure 5 shows only the study area (as is now more clearly stated in the caption), with the location of all the sites and symbol size referring to site area. Thus, the figure already includes what the reviewer asks for.

p. 10 (“... presence and intensity ...”) intensity being a sedentary society here?

We have replaced “intensity” with a following sentence stating that the formation of ADE suggests long occupation.

p. 10 (“... has been built inside ...”) can this be shown on a figure?

We now refer to Fig 4e as showing the location of a mound village.

p. 11 (“Medium to large...”) these are actually important results and should be clearly placed in a results section p. 11 (“ We interpret...”) same comment as previous - this is really the crux of the results - and it gets diluted in the text that contains so much site description and methods

We have moved the relevant text to the results section.

p. 11 (“Fig. 4e”) I would suggest drawing boxes around the areas of interest in fig 4e instead of drawing on top of the features, which make it impossible to see the actual features

We have now only outlined the features instead of drawing on top of them.

p. 12 (“We conducted test excavations...”) this section again jumps back to site descriptions (based on previous work) and methods for the current analysis, but is interspersed with results

I would suggest restructuring into clearly defined results/discussion and methods sections.

The site description and excavation are now presented in the Methods section. The results of the excavation have been moved to the Results section.

p. 13 (“We have modelled...”) it is unclear whether the model was used to predict sites that you tested with excavations or whether the excavations were sites included in the model

We have now stated that we built the model including all previous sites plus our sites surveyed in the UTB.

p. 14 (“... and distance from rivers ...”) can you report specific distances in comparison with the ADE results?

The maximum distance has been reported.

p. 15 (“... are yet to be found”) is this based on environmental similarity across the region or purely on geographic area? I think this should be clarified, especially given the following sentences

We have clarified that sentence by stating that it is a projection of the site density in the UTB to the total area predicted by the model in southern Amazonia.

p. 16 (“... sub-region of the SRA.”) I would suggest defining the sub-regions or maybe showing them on a figure

We have now labelled the sub-regions mentioned in the text in Fig. 1a.

p. 17 (“... basins of the Juruena ...”) is this the red box in figure 1a? please clarify

We refer to the appropriate figures now.

p. 17 (“...and circular mounded villages”) It is still a bit unclear in the manuscript whether this imagery was used to test the maxent model predictions, or whether it was used to provide occurrence locations to input into the model in the data-sparse region

We say in the methods section: “Ditched enclosure coordinates were compiled from published literature, unpublished theses and field reports, to which we added the newly discovered sites of the UTB, totalling 665 occurrence points”. Later, we say: “we performed ten partitions of the occurrence dataset, each time randomly selecting 75% of the sites as training sample and 25% as test sample”. Thus, it is clear now that we used all recorded sites + our new sites as the input, and randomly selected 25% of all those sites for the testing sample in ten iterations of the model.

p. 17 (“... according to area.”) can you define size classifications?

We have now reported the sizes used in the final classification, which was previously in the main text.

p. 17 (“In all cases...”) this is actually part of your results, and is quite interesting. i would suggest moving this text into your results section

We have moved this text to the results section.

p. 18 (“... 665 occurrence points.”) Food for thought here...it would be nice to have constructed the model using only the previously published occurrence locations from both the eastern and western sections. Then test the model using your imagery and newly detected sites. It would be a really cool way to show how the model performs in underrepresented locations, and an assessment of its robustness in certain conditions. If you don't want to reconfigure this manuscript, then I would suggest that we team up in the future and try an approach like this on a variety of site types and across a variety of regions.

We will certainly consider this suggestion for future collaborative work.

p. 18 (“We excluded...”) Here I think it would be good to define the area included in your study area - figure 1 looks to be defined as Amazonia sensu stricto (according to Eva et al. 2005) and defined as all regions draining into the Amazon River system and that are less than 1000 m elevation

We have added that definition to our description of the model area.

p. 18 (“ ... precipitation of warmest quarter ...”) in a manuscript that i currently have in review, i am getting criticized for using some of these variables - particularly ones that are hard to interpret ecologically. i agree with the reviewers, and will construct future models with a reduced set of predictor variables. maybe something to think about here as well?

We will take that into consideration for future models.

p. 18 (“ ... and spatial autocorrelation ...”) insert references here

We have inserted an appropriate reference.

p. 19 should kilometer be spelled out here?

We have spelled it out.

p. 19 (“In the evaluation...”) this is all results, and should be moved to the results section of the main text or of the supplementary material

The text has been moved to the results section.

p. 19 (“... and temperature seasonality.”) This bit of text is confusing. It would be better to have a table of those variable importances for the reader to refer to - like supplementary table 6.

We have now moved Supplementary Table 6 to the main text (as Table 2) and referred to it in this paragraph.

p. 19 how does it compare with the bioclim predictors? and p. 19 incomplete comparison here - in relation to which other variables? all of them? if it is the second-most important overall, then I think some of the text and interpretation should be revisited.

We have now stated that distance to rivers was the second most significant **among the terrain variables**, but that it was still low when compared to the bioclimatic ones.

p. 20 not needed since it was said in the previous sentence

The sentence has been removed.

p. 20 (“Population densities ...”) this is back to methods. I think the manuscript would be greatly improved by some restructuring of the methods section, which currently includes a substantial amount of results that are important to the main message of the paper

The sections have been restructured accordingly. The relevant text has been kept in the methods section.

p. 21 (“... the resulting area ...”) this is also results, and i would do a bit of restructuring in this section and p. 21 (“Assuming site contemporaneity...”) these are results and discussion that are central to the main message of the paper. I think these results should be presented outside of the methods section

The relevant paragraphs have been moved to the results section.

p. 28 (caption of Figure 1) clarify whether these are sites reported in previous literature or those being reported in this study.

define red box (study area of 1800km²???)

provide references for "other sites" in the caption"

We now describe the sites as “previously recorded”. The red box labelled UTB is now defined in the caption as comprising our research area, and it is specified that it is shown in detail in Figure 5. References to other sites have been added.